# ALTARED ENVIRONMENTS: THE ROLE OF NORMATIVE INFRASTRUCTURE IN AI ALIGNMENT

## ABSTRACT

Cooperation is central to human societies, which they achieve by constantly tackling the alignment problem of ensuring self-interested individuals act in ways that benefit the groups in which they live. As AI agents become pervasive in shared environments, it will be similarly crucial for them to align with the cooperative goals of human groups. Current AI alignment research largely focuses on embedding specified or learned norms into agents to achieve this cooperation. While valuable, this approach overlooks the role that institutions play in aligning human behavior to achieve cooperative gains and thus overlooks a potential alignment technique for AI agents. We address this gap by proposing **Altared Games**, a novel formal extension of Markov games that incorporates an *altar*—a classification institution providing explicit normative guidance to agents. Our approach focuses on a challenging setting where norms are dynamic, thereby requiring agents to adapt to the evolving norm content represented by the altar. Using multi-agent reinforcement learning (MARL) as a computational model of AI agents, we conduct experiments in two mixed-motive environments: Commons Harvest, which models resource sustainability, and Allelopathic Harvest, which involves coordination under conflicting incentives. Our results demonstrate that the altar enables agents to adapt effectively to dynamic norms, engage in accurate sanctioning, and achieve higher social welfare compared to systems without a classification institution. These findings highlight the importance of normative institutions in fostering cooperative, adaptable AI agents operating in complex real-world settings.

## 1 INTRODUCTION

The alignment challenge – how to ensure that self-interested individuals adopt behaviors that benefit the groups they live in – has been a persistent and evolving problem throughout human history. Efforts to address this challenge have been foundational for achieving the kind of ultra-cooperative societies humans have built. By engaging in schemes of task specialization, exchange, and mutual aid, which require individuals to follow group norms of appropriate behavior, humans have achieved levels of cooperation far beyond anything we see in other mammals Henrich (2016).

Integrating AI into human society extends the alignment challenge to artificial agents: how do we make sure these agents take actions that align with the norms of appropriate behavior that undergird our complex cooperative schemes? To date, this challenge has largely been framed in terms of how we embed values and norms into AI systems. And while this approach has been important for the safe deployment of existing systems, it is inherently limited. Human norms are dense–just about everything we say or do is subject to normative evaluation. Norms are dynamic, constantly adapting to changes in environments, populations, and information, and open to continuing contestation. And norms are highly differentiated: they range from ineffable standards such as how long it is appropriate to make eye contact with a stranger to legible norms about color code to at a funeral or how much food to take from a shared plate to formal legal requirements such as the obligation to take reasonable care while driving or to put away your garbage cans within 24 hours of collection. It is simply not possible to articulate all our values and norms Hadfield-Menell & Hadfield (2018).

Tackling the AI alignment challenge in a robust way will require taking fully on board the density, dynamism and differentiation of human norms and the lessons from how human groups throughout history have tackled the alignment challenge, successfully enough to have achieved extraordinary

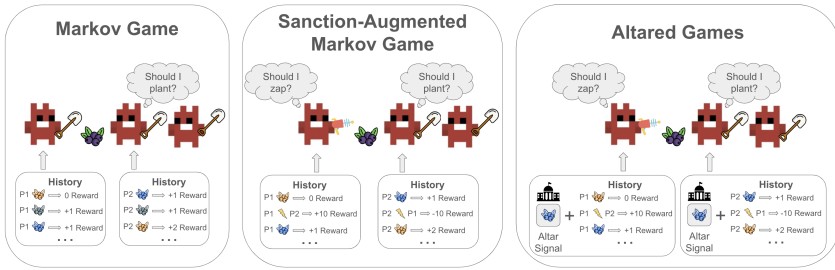

Figure 1: Overview of the **Altared Games**: Markov Game extension including third party enforcement mechanism and an environmental feature, called *altar*, encoding the norm that evolve over time. Learning unfolds in this order: Agents first learn to punish in accordance with the hidden reward structure for norms. In the presence of altar, they learn to map the observation to this hidden structure. This enables them to potentially predict the sanctioning behavior of other agents (a challenging problem) and then, as a consequence, they learn to avoid sanctions and comply with norms.

gains from cooperation. What we see in the human model is that humans do not, by and large, encode specific values and norms early in life that then guide lifelong behavior. Instead, human societies rely on constantly evolving interactions with other agents, normative signals, and, most fundamentally, *normative institutions* - authoritative common knowledge structures like groups of elders or courts that articulate, interpret and adapt norms–to align individual behaviors in dynamic environments and with dynamic populations. We can understand the emergence and evolution of normative institutions as a response to the density and dynamism of norms. Individuals in groups that lack an authoritative normative institution must extract information about the current norms and how they are being interpreted and enforced by other agents from agent behavior alone. Individuals in groups with an authoritative normative institution face a less computationally taxing and error-laden challenge in maintaining coordination of their enforcement and compliance behavior with the group. This generates group benefits in the form of increased social stability.

In this paper, we draw on this human model to investigate whether a normative institution can improve the capacity of architecturally simple AI agents to adapt to dynamic norms while solving social dilemmas Dawes & Messick (2000). We begin with the theoretical framework of (Hadfield & Weingast, 2012), which introduced a rational agent model of normative social order. In this framework, the challenge of inducing behaviors aligned with a group's norms resolves to the challenge of incentivizing and coordinating agents to punish in accordance with the normative classification (which behaviors are allowed, which are not allowed) articulated by a public classification institution. This framework provides a microfoundational account of the decentralized enforcement mechanisms seen in human societies, such as social disapproval or exclusion, that are a primary mechanism for incentivizing norm compliance.

We adapt this theoretical framework to the AI context with an extension to Markov games – a robust computational method for modeling sequential decision-making in multi-agent environments. We formalize decentralized enforcement in this setting by endowing agents in a multi-agent reinforcement learning (MARL) setting (following Perolat et al. (2017)) with a sanctioning technology by which they can deliver costly punishment to other agents. Following Köster et al. (2022), we implement norms by encoding rewards for agents that use their sanctioning technology to punish agents that have taken actions deemed by the norm to be punishable. If all or most agents reliably punish in this way, they also avoid violating the norm themselves to avoid punishment from others.

The punishment rewards in this framework are "hidden" in the sense that they are supplied by the environment and not modeled by the agents. How to earn rewards for punishing is thus a learning problem, one that (Köster et al., 2022) show MARL agents can solve with static norms/rewards. We make this learning problem harder and more realistic by implementing dynamic norms that change in a randomly controlled way during training. We then evaluate the impact of introducing a normative classification institution, a feature that provides a publicly observable representation of the current norm, that is, the current reward structure for punishment. We call this feature an **altar**, to capture the idea of an authoritative focal point in the environment that articulates a group's shared norms or laws, and propose a novel extension of the Markov game setting called **Altared Games**. Our research question then is: *does the introduction of an altar make it easier for agents to adapt their*

*punishment behaviors to changes in the norms and thus for a group of agents to maintain dynamic normative social order?*

To empirically investigate this framework, we conduct experiments in two mixed-motive games with different cooperative challenges: Commons Harvest, modeling resource sustainability under the tragedy of the commons; and Allelopathic Harvest, involving equilibrium selection and a free-riding challenge. To address our research question we use a methodology of controlled hypothesis testing to isolate the role of the altar, ensuring that the effects of the altar can be assessed independently by holding all other factors constant across experimental setups. Specifically, we train agents under three experimental conditions: (1) a *vanilla baseline* in which agents possess a sanctioning technology but sanctioning only generates private rewards (such as removing a competitor from a contested resource) meaning there are no pro-social rewards for sanctioning and hence no norms; (2) a *hidden-rule* environment in which agents are rewarded by the environment for punishing in accordance with the current norm; and (3) the *altared* environment which enriches the hidden-rule environment with an altar that represents the state of the current norm. In both the hidden-rule and the altared conditions the norms follow the same dynamic evolution and in both cases the group of agents would achieve the highest possible payoff if agents immediately shifted their behavior to align with the current norm. (Note that this means we are not in this paper investigating how a group identifies and adopts the optimal norms). Thus a comparison of performance between agents trained with a hidden-rule and with an altar isolates the impact of the altar alone. Our results show that the altar significantly improves agents' ability to adapt to dynamic norms, engage in correct sanctioning behaviors, and achieve higher social welfare efficiently, even under uncertainty.

In sum, our work provides a first step toward understanding how normative institutions can enhance alignment in multi-agent systems. Focusing on dynamic norms and explicit institutional guidance, we aim to pave the way for future research into scalable, adaptable, and socially aligned AI systems.

## 2 PRELIMINARIES

This section establishes the theoretical and formal foundations of this work. We summarize the theoretical framework of normativity, focusing on the role of classification institutions and enforcement mechanisms in sustaining cooperation. We then describe Markov games and its extension to sanction-augmented Markov games, which incorporate third-party enforcement into a computationally rich multi-agent setting. An extended related work discussion is available in Appendix B

### 2.1 THEORETICAL FRAMEWORK: HADFIELD-WEINGAST MODEL

Our investigation is grounded in a parsimonious rational agent model of normative social order introduced by Hadfield and Weingast (Hadfield & Weingast, 2012). This model provides a structured perspective on how groups sustain cooperation by leveraging two essential components: a classification institution and an enforcement mechanism. The classification institution provides common knowledge binary classifications of behaviors as either "punishable" or "not punishable," potentially through the application of general principles to specific cases. These classifications reduce ambiguity, creating a shared understanding of acceptable behavior within the group. The enforcement mechanism incentivizes agents to align with these classifications by imposing penalties on punishable actions, encouraging agents to favor "not punishable" behaviors. A stable normative social order is achieved when most agents are mostly in compliance and avoiding punishment.

(Hadfield & Weingast, 2012) focus in particular on the case, which describes most of human history and much of modern life as well, in which punishment is decentralized, that is, primarily delivered by ordinary agents (rather than specialized enforcers Hadfield & Weingast (2013).) Agents must therefore be incentivized and coordinated to engage in costly third-party punishment (which could be relatively mild, such as criticism, or more harsh, such as exclusion from the group) and to condition such punishment actions on a shared classification institution..

Although shared classification could be entirely emergent and informal[1], groups that converge on a single authoritative (more formal) classification institution–such as a chief, a group of elders,

---

[1]There is no entity that tells members of the group that they should honk at a car that is failing to facilitate merging on the highway but everyone in the group could reliably say that this is the norm Bicchieri (2005)

or a court–can enjoy significant benefits. These formalized systems help resolve ambiguities and improve normative clarity, provide consistency and help maintain cooperation even in the face of dynamic environments and populations (Hadfield, 2017).

## 2.2 MARKOV GAMES

**Markov games**, also known as stochastic games, extend Markov decision processes to multi-agent settings, providing a general framework for modeling dynamic interactions among agents. A Markov game is defined as a tuple: $\langle S, A, P, R, \gamma, n \rangle$, where, $S$ is the shared state space, representing all possible configurations of the environment; $A = A_1 \times A_2 \times \cdots \times A_n$ is the joint action space, where $A_i$ denotes the set of actions available to agent $i$, and $n$ is the total number of agents; $P(s' \mid s, a)$ is the transition function, specifying the probability of transitioning to state $s'$ from state $s$ given the joint action $a$; $R = (R_1, R_2, \ldots, R_n)$ represents the reward functions for each agent, where $R_i(s, a)$ determines the reward received by agent $i$ after taking action $a$ in state $s$; $\gamma \in [0, 1)$ is the discount factor, controlling the relative importance of future rewards.

In **partially observable** settings, each agent $i$ does not have access to the full state $s$ but instead receives an observation $o_i \in O_i$, drawn from the observation function $O_i(s)$. The Markov game is extended to a partially observable Markov game (POMG) by redefining the tuple as: $\langle S, A, P, R, \gamma, n, O \rangle$, where $O = (O_1, O_2, \ldots, O_n)$ defines the observation spaces of the agents. The addition of partial observability introduces complexity, as agents must infer hidden state information from their observations to make optimal decisions. At each timestep, agents observe $s$ (or $o_i$ in the partially observable case), select actions $a_i \in A_i$, and transition to a new state $s'$ based on $P(s' \mid s, a)$. Each agent's goal is to learn a policy $\pi_i : S \to A_i$ (or $\pi_i : O_i \to A_i$ in the partially observable case) that maximizes its expected cumulative discounted reward: $\pi_i^* = \arg\max_{\pi_i} \mathbb{E}\left[\sum_{t=0}^{\infty} \gamma^t R_i(s_t, a_t)\right]$. A subset of Markov games, **mixed-motive settings**, involves a combination of cooperative and competitive incentives. These settings model scenarios where agents must balance individual objectives with the collective good, often facing dilemmas such as the equilibrium selection problem, where multiple equilibria, generally with different individual and aggregate payoffs, exist; the free-rider problem, where agents benefit from shared resources without contributing to their production or maintenance; or the tragedy of the commons, where uncoordinated actions lead to the depletion of shared resources. Studying such settings is central to our investigation, as they highlight the challenges of aligning individual incentives with group goals and provide a rich domain for exploring the role of norms and enforcement mechanisms.

**Multi-Agent Reinforcement Learning (MARL)** provides the computational framework for solving Markov games, where agents interact with the environment and each other to optimize their policies. Formally, each agent $i$ learns a policy $\pi_i$ to maximize its cumulative discounted reward: $\pi_i^* = \arg\max_{\pi_i} \mathbb{E}\left[\sum_{t=0}^{\infty} \gamma^t R_i(s_t, a_t) \mid \pi_1, \ldots, \pi_n\right]$, where $a_t = (a_{1,t}, a_{2,t}, \ldots, a_{n,t})$ represents the joint action at timestep $t$. MARL approaches can involve optimizing a joint policy $\pi = (\pi_1, \ldots, \pi_n)$ under shared information or decentralized policies where agents act independently. For this work, we consider a simple, practical and scalable approach to MARL is Independent Proximal Policy Optimization (IPPO) de Witt et al. (2020a), a decentralized method where each agent optimizes its policy independently using a variant of Proximal Policy Optimization (PPO).

## 2.3 SANCTION-AUGMENTED MARKOV GAMES (SMG)

We consider an extension of Markov games to include sanctions (punishment), called **Sanction-Augmented Markov Games (SMG)**. This framework formalizes how agents can impose penalties on others to enforce compliance with norms. An SMG is defined as: $\langle S, A', P, R, \gamma, n, \Sigma, C, I \rangle$, where $S, A, P, R, \gamma, n$ retain their meanings from the standard Markov game definition. Each agent $i$ has an extended action space $A_i' = A_i \cup \{\sigma\}$, where $\sigma$ is a common sanctioning action (e.g. zapping or criticism) available to all agents. The transition dynamics $P(s' \mid s, a)$ determine the next state $s'$, while the reward functions $R = (R_1, R_2, \ldots, R_n)$ incorporate the effects of sanctions into individual incentives. This form of SMG (not including C and I, defined below) was introduced in Perolat et al. (2017) where they endowed agents with a punishment (sanctioning) technology.

Köster et al. (2022) took this extension a step further by implementing a **hidden classification rule** that implemented norms by rewarding agents for sanctioning behaviors designated exogenously (i.e. by the researchers) as norm violations. Formally, the sanction cost function $C = (C_i, C_j)$ defines

the costs associated with sanctions: $C_i(\sigma) = p_i$, the cost incurred by the sanctioning agent $i$, and $C_j(\sigma) = p_j$, the cost (penalty) incurred by the sanctioned agent $j$. The indicator variable then implements a classification of actions as either norm violations or not: $I_j(s, a) = 1$ if the action $a$ taken previously by agent $j$ violates a norm, and $I_j(s, a) = 0$ otherwise. To incentivize sanctioning of norm violations, the sanctioning agent receives a reward $q$ that offsets the cost $p_i$, resulting in a net positive reward ($q - p_i$) if the sanctioned agent $j$ violated a norm ($I_j(s, a) = 1$) in the prior step. We call sanctioning of designated norm violations *correct sanctioning*. Conversely, if the sanctioned agent did not violate the norm ($I_j(s, a) = 0$), the sanctioning agent incurs the full cost of sanctioning $p_i$ without any offsetting reward. We call this *incorrect sanctioning*. Formally, the reward for the sanctioning agent $i$ is: $R_i(s, a, \sigma) = R_i(s, a) - C_i(\sigma) + I_j(s, a) \cdot q$. For the sanctioned agent $j$, the reward function reflects the penalty for being sanctioned, irrespective of whether the agent violated a norm: $R_j(s, a, \sigma) = R_j(s, a) - C_j(\sigma)$. This formalization captures both the costs and rewards of sanctions, emphasizing the role of accuracy and cost-effectiveness in enforcement. Sanctioning agents are incentivized to sanction correctly to offset their costs, while incorrect sanctions lead to a net penalty. Sanction-augmented environments align well with the (Hadfield & Weingast, 2012) theoretical framework, modeling third-party enforcement to secure normative social order.

## 3 OUR APPROACH: ALTARED GAMES

As discussed in 2.3, (Köster et al., 2022) leveraged the SMG framework for introducing a hidden classification rule, rewarding agents for sanctioning behaviors aligned with predefined but implicit norms. They demonstrated that MARL agents can efficiently learn to punish, and therefore comply, with researcher-set norms. In this paper we extend this framework in two new ways:

First, we address the challenge of **dynamic norms**, where the classification behaviors as punishable or acceptable evolves over time. Second, we introduce a **normative institution**, called the **altar**, which encodes the prevailing norms in the environment. The altar is implemented as an observational feature of the environment and does not modify the structure of the underlying SMG. The altar makes normative content legible to agents. We hypothesize that as a result of this enrichment of the environment, the altar facilitates agent learning and coordination in environments with dynamic norms relative to the hidden rules environment studied by (Köster et al., 2022).

### 3.1 DYNAMIC NORMS

In the context of Sanction-Augmented Markov Games (SMGs), we formalize dynamic norms as a time-dependent mapping: $N_t : S \to \mathcal{A}$, where $N_t(s) \subseteq \mathcal{A}$ defines the set of acceptable (not punishable) actions in state $s$ at time $t$. The evolution of norms is governed by an update function: $N_{t+1} = f(N_t, \Phi)$, where $f$ captures the mechanism of norm evolution, and $\Phi$ represents triggers or drivers of change. We do not model the determinants of norm evolution but these drivers could be thought of as arising from external inputs (e.g., regulatory updates or environmental changes), agent-driven mechanisms (e.g., collective decision-making or voting), or stochastic events (e.g., resource depletion or unexpected disturbances).

Dynamic norms pose two main challenges for agents. First, dynamic norms require agents to continuously track the norm as it evolves, updating their internal models based on observed rewards, sanctions, and environmental cues. Second, agents must adjust their strategies to align with shifting expectations about rewards while navigating a mixed-motive setting. In our setup, these challenges are particularly acute. The immediate impact of a change in the norm is not on the rewards associated with actions that either comply or not with the norm; rather, it is on the rewards associated with sanctioning actions. The impact of norm change on compliance is only derivative: *if* enough agents adapt their sanctioning behaviors to accord with the new norm, then agents will adapt their compliance behaviors to accord with the new norm. ((Köster et al., 2022) show that this learning process is sequential: MARL agents first learn to punish in accordance with the hidden reward structure for norms and then, as a consequence, they learn to comply with norms. Predicting the rewards associated with compliance and non-compliance (which can impact individual payoffs as norms in a mixed-motive setting generally will sometimes require agents to forego self-interested actions in favor of pro-social actions), then is a very difficult problem in a multi-agent setting as it requires predicting the enforcement behavior of other agents.

Real-world examples include resource management scenarios, where norms shift in response to scarcity, and traffic systems, where acceptable behaviors adapt to changing infrastructure or population density. By incorporating dynamic norms into SMG, we aim to model these complexities and investigate how agents operate in environments with evolving expectations.

## 3.2 ALTARED GAMES

Our aim is to test the value of a normative institution, which we call the altar and which encodes the prevailing reward structure for punishment (the norms). We thus further extend the SMGs framework to incorporate the altar feature and call this *Altared Sanction-Augmented Markov Games (Altared SMGs)*, which we will refer to as **Altared Games** for short. An Altared SMG is defined as: $\langle S, A', P, R, \gamma, n, \Sigma, C, I, \mathcal{M}_{\text{altar}} \rangle$, where $S, A', P, R, \gamma, n, \Sigma, C$, and $I$ retain their meanings from the SMG framework, and $\mathcal{M}_{\text{altar}} : S \times \mathcal{A} \to \{0, 1\}$ is a mapping function managed by the environment, encoding the normative classification of actions. It specifies whether an action $a$ in state $s$ complies with the norm, with $\mathcal{M}_{\text{altar}}(s, a) = 1$ indicating compliance and $\mathcal{M}_{\text{altar}}(s, a) = 0$ indicating violation. Combined with the sanctioning mechanism described above, the altar thus encodes the sanctioning reward structure, indicating what constitutes correct sanctioning.

Agents do not have direct access to $\mathcal{M}_{\text{altar}}$. Instead, when visiting a designated subset of states $S_{\text{altar}} \subseteq S$, they receive an observation $o_{\text{altar}}$ that implicitly reflects the normative content encoded by $\mathcal{M}_{\text{altar}}$. The indicator variable $I_j(s, a)$, which specifies whether the action $a$ by agent $j$ violates the norm, is derived implicitly from $\mathcal{M}_{\text{altar}}$: $I_j(s, a) = 1 - \mathcal{M}_{\text{altar}}(s, a)$. Thus, while $\mathcal{M}_{\text{altar}}$ governs the normative structure of the environment, agents must infer this structure through observation and feedback. Moreover, the reward functions for the sanctioning agent $i$, $R_i(s, a, \sigma)$ and the sanctioned agent $j$, $R_j(s, a, \sigma)$ remain consistent with the SMG framework. The altar observations merely provide agents with additional information about the rewards for sanctioning. Thus, while there is no direct cost associated with visiting $S_{\text{altar}}$, interactions with the altar involve implicit opportunity costs: Agents forgo potential reward-generating actions during the time spent visiting $S_{\text{altar}}$.

This formalization bridges the gap between implicit norm enforcement in hidden rule systems and explicit norm representation. By incorporating the altar into the SMG framework, we create a testbed for investigating how observable classification institutions influence agent learning, coordination, and compliance in dynamic, multi-agent environments.

## 4 EXPERIMENTS

The objective of our experiments is to evaluate the impact of the altar, on agent behavior, norm learning, enforcement, and compliance in dynamic multi-agent systems. Specifically, we aim to understand whether making norms observable through the altar improves agents' ability to align with evolving norms, enforce compliance, and achieve higher overall system efficiency compared to configurations without explicit institutional representation.

To explore these questions, we use two mixed-motive environments: **Commons Harvest** Perolat et al. (2017) and **Allelopathic Harvest** Köster et al. (2020). To realize these environments, we leverage the Melting Pot Suite Agapiou et al. (2023); Leibo et al. (2021), a flexible research platform that provides high-fidelity multi-agent environments with diverse incentive structures and interdependencies. Its extensibility allows us to adapt these environments systematically to include explicit institutional mechanisms like the altar. The aim of our experiment is to test the hypothesis that the presence of an altar that encodes norms improves the capacity of agents to implement norms. For this reason, following (Köster et al., 2022), we exogenously control the content of norms. We test our hypothesis by training agents under three experimental conditions. In the **Vanilla** Baseline (Markov Game), the original Markov game is used without norms or sanctioning technology, and agents maximize individual rewards without external guidance or sanctions. This gives us a reference point to assess the group benefits achieved if the agents are able to implement our deliberately group-beneficial norms. Our second condition, **Hidden Rule SMG**, introduces the sanctioning technology and the hidden reward structure that rewards sanctioning according to the current norms. Finally, the **Altared SMG** condition incorporates the altar but is otherwise the same as the Hidden Rule SMG condition, with dynamic norms that follow the same evolution and the same rewards for

punishing according to these norms. By comparing performance for agents trained under these three conditions, we aim to isolate the effects of the altar on agent behavior and system outcomes.

## 4.1 Environments: Core, SMG and Altared Versions

In this section, we provide an overview of the two environments used in our experiments: **Commons Harvest** and **Allelopathic Harvest**. For each environment, we first outline the core mechanics, describing the resource dynamics and agent interactions that define the setting. We then explain how the Sanction-Augmented Markov Game (SMG) version of the environment is constructed, leading to the **hidden rule** mechanism for enforcing norms. Finally, we detail the steps taken to convert these environments into their **Altared** versions, explicitly incorporating the altar as an observable institution encoding norms.

### 4.1.1 Commons Harvest[2].

**Core Mechanics.** In this environment, agents aim to collect apples scattered across six distinct patches, earning a reward of +1 for each apple consumed. Apple regrowth depends on the density of neighboring apples within a Euclidean radius of 2, with probabilities decreasing as local density declines: 0.025 for three or more neighbors, 0.005 for two, 0.001 for one, and 0 for none. Overharvesting a patch depletes it permanently, requiring agents to reduce collection to sustain resources. A social dilemma ensues: consuming the last apple in a patch generates individual rewards but risks permanent patch depletion, leading to the tragedy of the commons.

For this work, we divide the six patches into three zones: the top two patches are designated red, the middle two are blue, and the bottom two are green. Agents are initially gray, but take on the color of the zone from which they consume apples. This color change makes their collection behavior observable by other agents. This setup lays the groundwork for introducing sanctioning based on zones in subsequent versions. We then train agents in three conditions described in Appendix A.1.

Achieving normative alignment in this environment translates to agents adapting to evolving norms and sanctioning other agents correctly. Because the norm is adjusted to reflect the current supply of apples across different patches, correct sanctioning in accordance with current norms incentives agents to adapt their harvesting behavior to the health of apple supply, mitigate overharvesting and thereby achieving higher collective welfare over time.

### 4.1.2 Allelopathic Harvest[3]

**Core Mechanics:** This environment poses both the coordination and the free-rider problem, making it challenging for agents to reach a welfare maximizing outcome. Specifically, in this environment, there are berries of three different colors and sixteen agents can plant and consume berries. Agents get reward for consuming any colored berry (+1) but receive higher reward for consuming their preferred color berry (+2). Planting does not generate any reward or cost and hence agents have no direct incentive to plant., leading to a free-rider problem. The agents can only consume ripened berries and the berry ripening rate is directly proportional to the fraction of the largest amount of berry color. Hence, if all three colors are equally distributed, berries will have the slowest ripening rate and achieving a monoculture of a single berry color will generate the highest berry ripening rate, thereby giving a chance to agents to accumulate more reward (equilibrium selection problem). Agents are initially gray, but take on the color of the berry they plant. This color change makes their collection behavior observable by other agents. This setup lays the groundwork for introducing sanctioning based on berry color for which monoculture is desired. We then train agents in three conditions discussed in Appendix A.2.

Achieving normative alignment in this environment translates to agents adapting to evolving norms and sanctioning other agents correctly. Because the norm is adjusted to reflect the currently desired

---

[2] Perolat et al. (2017) introduced this environment to investigate the ability of multi-agent reinforcement learning agents to coordinate in solving common-pool resource appropriation problems, building on the mechanics first outlined in Janssen et al. (2010)

[3] Köster et al. (2020) introduced this environment to investigate the ability of multi-agent reinforcement learning agents to overcome free-rider problem while solving equilibrium selection problem rooted in the allelopathic mechanic, previously studied in in Leibo et al. (2019)

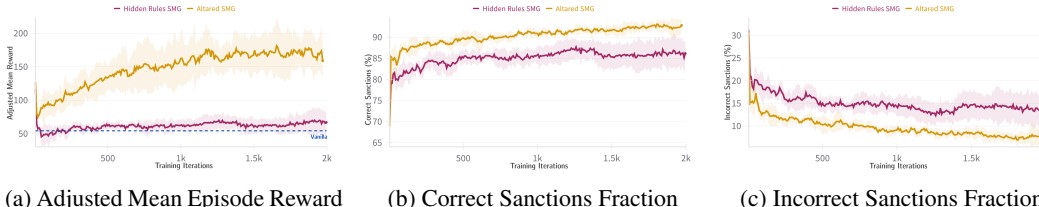

(a) Adjusted Mean Episode Reward     (b) Correct Sanctions Fraction     (c) Incorrect Sanctions Fraction

Figure 2: Results on Altared Commons Harvest: Adjusted reward mean discounts the reward obtained for sanctioning. All experiments run for 5 seeds.

monoculture color, correct sanctioning in accordance with current norms incentives agents to adapt their planting behavior to the desired monoculture while avoiding to free-ride, thereby achieving higher collective welfare over time.

## 4.2 RESULTS

Our empirical investigation focuses on assessing the impact of the altar on agents' capacity to implement norms, compared to environments without the altar. During training, we expect the learning process to unfold as follows: agents first learn to recognize nonacceptable behaviors by receiving rewards for sanctioning violations, enabling them to enforce punishments correctly. Over time, this enforcement leads agents to predict actions likely to result in sanctions, prompting them to learn compliant behavior by avoiding such actions. This progression drives agents toward maintaining a normative social order. In the presence of the altar, agents would be required to visit it periodically to update their understanding of the prevailing norm. They must learn to map altar observations to appropriate sanctioning behaviors, potentially facilitating faster and more accurate adaptation to changing norms. We present our results through the lens of this learning process, comparing agent performance in environments with and without the altar (everything else being the same) at each stage of this progression. The results are reported over 5 seeds for all experiments and further training details are available in Appendix C

We highlight that agents not engaged in a normative system face prohibitive difficulty in learning the restraint required in Commons Harvest to avoid the tragedy of the commons, as observed in agents trained under the vanilla condition. In the Allelopathic Harvest environment, these agents tend to free-ride by consuming berries indiscriminately, preventing any increase in the growth rate of berries and resulting in stagnation at a specific reward level.

**Agents learn correct sanctioning behavior in the presence of an altar.** As a first result, we measure the impact of institution on the ability of agents to enforce punishments correctly. For this, we plot the fraction of the correct and incorrect sanctions that agents engage in over the course of training for the baseline without institution and our approach. Figures 2b, 2c and Figures 3b, 3c shows the results for Common Harvest and Allelopathic Harvest environments respectively. In the Commons Harvest environment, agents trained in the Altared SMG framework quickly learn to perform the majority of their sanctions correctly and adapt to dynamic norms. They maintain a high fraction of correct sanctioning behavior over extended training periods while significantly reducing variance compared to the Hidden Rule SMG baseline. This highlights the advantage provided by the altar feature in facilitating agents' understanding of the sanctioning reward structure. In the Allelopathic Harvest environment, agents trained in Altared SMG initially struggle to identify correct sanctioning behaviors. However, they eventually match the performance of the Hidden Rule SMG baseline and, over time, appear to surpass it. Additionally, the variance in performance is consistently lower for Altared SMG compared to Hidden Rule SMG, underscoring the stabilizing effect of explicit institutional guidance.

**Agents learn to visit the altar consistently.** In the challenging setup of dynamic norms, it is crucial for agents to learn to visit the altar at regular intervals to stay updated on the evolving normative content. To evaluate this behavior, we tracked the number of visits made by agents to the altar over the course of training. Figures 4b and 4a depict the visitation patterns for the Commons Harvest and Allelopathic Harvest environments, respectively. Our results show that, over time, agents in both environments converge on a consistent visitation pattern, maintaining a stable frequency of visits to

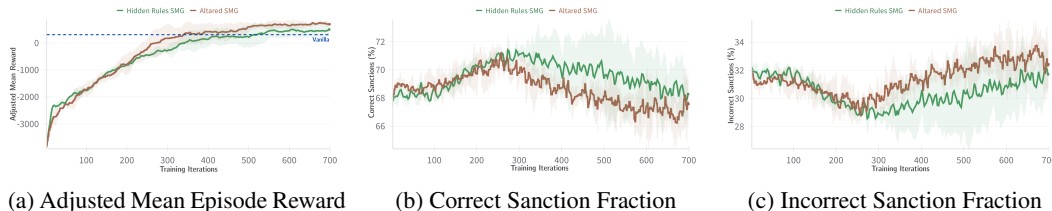

(a) Adjusted Mean Episode Reward     (b) Correct Sanction Fraction     (c) Incorrect Sanction Fraction

Figure 3: Results on Altared Allelopathic Harvest: Hidden Rule SMG has high variance compared to Altared SMG, which also achieves 200 points higher reward. All experiments run for 5 seeds.

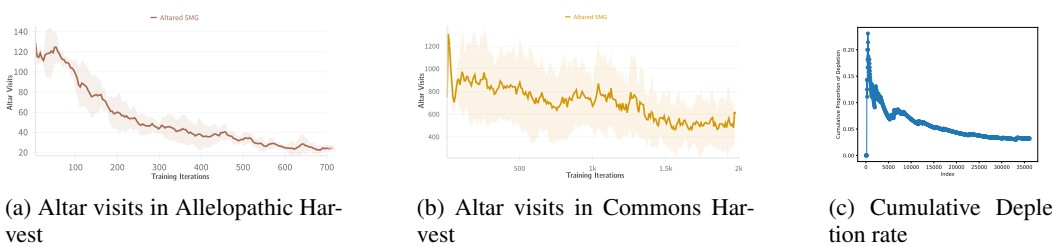

(a) Altar visits in Allelopathic Harvest     (b) Altar visits in Commons Harvest     (c) Cumulative Depletion rate

Figure 4: Results on Altar visits in both environments and Depletion rate in Commons Harvest

the altar. This behavior suggests that agents effectively learn the importance of periodic updates from the altar to adapt to the dynamic norms.

**Agents obtain high collective welfare, more efficiently when trained in the presence of altar.** It is important to note that agents in both the Altared SMG and Hidden Rule SMG setups receive rewards for correctly sanctioning violations. However, this reward is artificial and is solely intended to train agents to learn proper sanctioning behavior. Sanctioning is inherently costly for both the source and target agents, and it is only justified if the rewards obtained from the base environment outweigh the associated costs. To account for this, we report the **Adjusted Mean Reward** in our results, which excludes the rewards earned from correct sanctioning. This metric ensures a fair evaluation of overall performance by focusing on the net benefits derived from the base environment while still incorporating the costs associated with sanctioning.

As shown in Figure 2a, Altared SMG demonstrates strong performance in Commons Harvest environment, where agents attain higher reward quickly and are able to sustain the increase in their collective reward. This requires the agents to show collective restraint towards harvesting apples from the zones that are nearing depletion. Surprisingly, the agents in the strong Hidden Rule SMG are not able to learn to show restraint and are not able to attain higher reward. It is important to note that the only difference between the baseline and our work is the presence of the institution in the environment. The rationale behind this performance is the crux of our position that institutions are important tools to achieve collective alignment – institutions take away the burden of enormous amount of computation required by the agents in order to understand the normative social order and reason about it. This effect results in the overall reduction in coordination costs, thereby im-

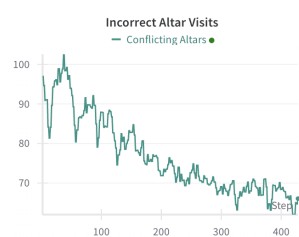

Figure 5: Visits to the altars representing content not correlated with sanctioning rewards

proving the efficiency of achieving cooperative outcomes. Qualitatively, one of the most important measure of success in Commons Harvest environment is the ability of the agents to foster sustainability. To assess this, we compute the cumulative depletion rate of apples that the agents cause over the training time. Figure 4c showcases the ability of agents trained in presence of altar to significantly reduce the depletion rate.

In the Allelopathic Harvest, Figure 3a similarly good results, albeit only marginally better than the Hidden Rule SMG baseline. We note that while the difference looks small in the plot, the

Altared SMG is better by a score of 200 points and with much less less variance compared to the Hidden Rule SMG ($> 200$ vs 60). This demonstrates that the institution already provides the necessary information to learn to attain and maintain higher welfare quicker and more reliably. In the Appendix D, we discuss a non-dynamic version of this environment to perform several qualitative analysis tasks which gives more insights into the behavior of agents in the presence of altar in this environment. Altared SMG's marginal performance improvement over Hidden Rule SMG in the current results can be attributed to the artifact that the training runs on Allelopathic harvest have not finished and it appears that agents have just begun to learn correct sanctioning behavior which may be the limiting factor.

**Altar provides value in terms of system robustness as agents learn to respond to different institutional configurations**

To address this question, we consider two variations over the basic version of Altared Allelopathic Harvest environment focusing on the robustness of the system when faced with different institutional configurations. More details on these environments are available in Appendix E.1.2.

**Alleopathic Harvest Limited.** This has same altar dynamics as Alleopathic Harvest Altar except that the institution is visible to the agents for only few time steps after the color change mimicking how some institutions are only accessible at particular times. In our results, we observe that the agents start visiting the institution during the first half of the interval since the color changes, thereby continuing to continue learning to adapt the dynamics institutional content despite limited visibility.

**Allelopathic Harvest Conflict.** This has two extra altars in the environment which serves as distractors to the agent. These altar will display information that do not align with the central altar. But the central altar is the one with the correct prescription and hence the agents need to learn to decrease their visit to the incorrect altar to be able to keep improving their ability to cooperate. Figure 5 demonstrates that the agents indeed learn to cut down their visits to the incorrect altar significantly and keep improving their overall reward, thereby responding and adapting to the correct institution.

## 5 DISCUSSIONS AND CONCLUDING REMARKS

In this work, we draw on the model of human societies to investigate whether a normative institution can improve the capacity of architecturally simple AI agents to adapt to dynamic norms while solving social dilemmas. Building on the theory of rational agent model, we propose a formal extension of Markov games, called Altared Games, which focuses on the decentralized enforcement mechanisms in multi-agent systems and introduces a feature called an **altar**, hat provides a publicly observable representation of the current norm, that is, the current reward structure for punishment. Using multi-agent reinforcement learning, we examine whether the introduction of an altar make it easier for agents to adapt their punishment behaviors to changes in the norms and thus for a group of agents to maintain dynamic normative social order. In a modified Allelopathic Harvest game and Commons Harvest game, we perform a controlled hypothesis driven testing and demonstrate superior performance of agents trained with an altar compared to those without it. While Altared Games have a strong theoretical grounding, it is also highly intuitive - institutions reduce the cognitive burden on agents when addressing cooperation challenges by tracking key elements that sustain and promote normative social order. This enables agents to efficiently engage in coordinated, cooperative behaviors, facilitating quicker and more effective collective action. This work provides a first step toward understanding how normative institutions can enhance alignment in multi-agent systems. By focusing on dynamic norms and incorporating explicit institutional guidance, we aim to pave the way for future research into scalable, adaptable, and socially aligned AI systems

Our current exposition provides a recipe for designing environments and systems with normative infrastructure as a key component, we strongly believe that this direction is ripe with immediate future avenues. We posit that our approach will be particularly effective in promoting generalization, adaptability, and robustness across environments with different normative institutions. Agents that learn to recognize altar information and correlate enforcement behaviors with norm content will adapt and train quickly when transferred to new environments. Further, normative institutions will be most effective in achieving collective alignment at scale when agent groups are large, as they provide structured guidance and shared norms that simplify decision-making and coordination across many individuals. This reduces the complexity of aligning diverse actions and fosters widespread cooperation, even in expansive groups.

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

# A DESCRIPTION OF ENVIRONMENT CONDITIONS

## A.1 COMMONS HARVEST

**Vanilla**: This environment retains the basic tagging (sanctioning) mechanism introduced by (Perolat et al., 2017). Tagging is costless for the sanctioning agent (other than opportunity cost) and removes the target agent from the environment for 25 steps. The tagged agent thus loses the opportunity to collect apples and the tagging agents benefits (if at all) from removing a competitor. This version establishes a baseline for agent behaviors and group performance in the absence of norms, when sanctioning can only generate private benefits for the sanctioning agent.

**Hidden Rule SMG**: This version incorporates pro-social rewards for sanctioning in accordance with a norm. The norm prescribes which zone it is acceptable for agents to harvest from at a given point in time. Initially, the acceptable zone is the one with the highest minimum apple count across its two patches. This prescription changes dynamically: when one of the patches in the acceptable zone falls below a threshold (four in our case) apples, the norm shifts to a zone with the highest minimum count that meets the threshold. If no such zone exists, no zone is prescribed, and harvesting is prohibited until regeneration occurs. This dynamic norm evolution ensures that acceptable behaviors adapt to resource availability. We call this the 'hidden rule' condition because the norms/rewards for sanctioning are generated by the environment and can only be discovered through sanctioning.

The tagging mechanism from the vanilla version is modified to enforce this normative structure. As before, tagging costs the tagging agent -10 and tagged agents are removed from the environment for 25 timesteps, losing harvesting opportunities. However, in addition to the private benefits to tagging experienced in the vanilla version, if the target of sanctioning violated the current norm by harvesting from a zone other than the one prescribed by the hidden reward structure, the tagging agent receives a reward of +20, resulting in a net reward of 10. This incentivizes agents to learn how to correctly sanction, which is a dynamic problem.

**Altared SMG:** This condition implements the same rewards for sanctioning and the same evolution of norms as the hidden rule condition but also includes an altar – an observable classification institution incorporated at three distinct locations within the environment, as shown in the Figure 6. It serves as an environmental feature encoding the currently prescribed norm. When agents visit an altar location, they receive an observation of its color, which matches the color of the zone from which it is currently acceptable to collect apples. If no zone is acceptable (i.e., all zones have insufficient resources), the altar displays a yellow fire symbol. The sanctioning mechanism remains consistent with the Hidden Rule SMG setup: agents earn rewards for tagging agents who collect from any zone other than the one prescribed by the altar (or

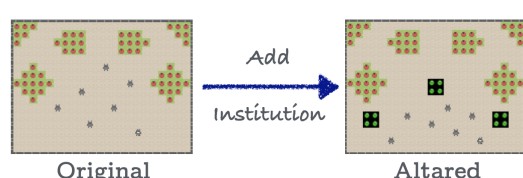

Figure 6: **Altared Commons Harvest**: Altars display the color of the zone from which is it currently acceptable to harvest. Three zones: red (top two zones, 6 apples each), blue (middle two zones, 10 apples each), and green (bottom two zones, 13 apples each). Altar displays green: indicating that the bottom zone is acceptable for harvesting. The altar displays yellow fire when harvesting from all zones is prohibited.

from any zone when all zones are prohibited.) Unlike the hidden rule condition, where agents can learn norms only from sanctioning and being sanctioned, in the Altared SMG the agents can also learn to recognize the altar, to visit the altar, and to map its observations to appropriate sanctioning behavior.

## A.2 ALLELOPATHIC HARVEST

**Vanilla:** This environment simplifies the two layer zapping (sanctioning) mechanism introduced by (Köster et al., 2020). Sanctioning is costless for the sanctioning agent (other than opportunity cost) and incurs a penalty of -10 for sanctioned agent. This version establishes a baseline for agent behaviors and group performance in the absence of norms, when sanctioning can only generate private benefits for the sanctioning agent.

**Hidden Rule SMG**: As before, this version incorporates pro-social rewards for sanctioning in accordance with a norm. The norm prescribes which monoculture is desired and planting the berry of that color is acceptable action. This prescription changes dynamically: in the episode of 2000 steps, we change the norm randomly every 100 steps for first 1000 steps and then 3-5 times at random interval (minimum gap of 200 steps between change). This is to ensure that agents get enough experience for each color, while also mimicking real world processes such as regulatory updates. The zapping mechanism from the vanilla version is modified to enforce this normative structure. As before, zapping costs the zapping agent -10 and tagged agents incurs cost of -10. However, in addition to the private benefits to tagging experienced in the vanilla version, if the target of sanctioning violated the current norm by planting the berry with color other than the one prescribed by the hidden reward structure, the tagging agent receives a reward of +20, resulting in a net reward of 10.

**Altared SMG**: This condition implements the same rewards for sanctioning and the same evolution of norms as the hidden rule condition but also includes an altar – an observable classification institution incorporated at the center of the environment, as shown in the Figure 7. When agents visit an altar location, they receive an observation of its color, which matches the desired berry color to be planted. This will include sanctioning gray agents too, thereby helping towards solving fre-rider problem. The sanctioning mechanism remains consistent with the Hidden Rule SMG setup: agents earn rewards for sanctioning agents who plant berry other than the one prescribed by the altar.

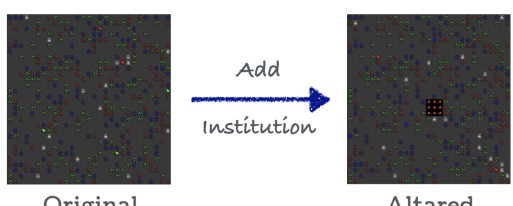

Figure 7: **Altared Commons Harvest**: Altars display the color of the berry for which monoculture is desired. Three colors: red, blue and green. Altar displays red: indicating that the red berry should be planted. Planting any other berry or free-riding is sanctionable.

## B EXTENDED RELATED WORK

There is a vast body of literature addressing various aspects related to our agenda, reflecting extensive efforts across multiple directions. To provide a clear understanding of the existing work, we categorize the related efforts into several key topics, which are discussed in detail below.

**Learning to cooperate in multi-agent systems.** The problem of cooperation—how to design environments and algorithms that align agents' behavior towards higher collective welfare—has received increasing attention in the multi-agent literature (Du et al., 2023). Common approaches include designing agents that have other regarding preferences through intrinsic rewards that promote collective welfare Peysakhovich & Lerer (2017) or acting altruistically toward others (McKee et al., 2020). Other methods use social influence Jaques et al. (2019) as an underlying mechanism; although these approaches are generally designed for coordination problems rather than cooperation. It is worth noting that in multi-agent reinforcement learning (MARL), the distinction between coordination and cooperation challenges is often blurred. Recently, norms have emerged as another set of mechanisms in MARL specifically designed to address cooperation. These methods draw on the extensive literature regarding the evolution of cooperation in human societies (Boyd & Richerson, 1992). By extending Markov decision processes (MDPs) with sanctions (Vinitsky et al., 2023), agent societies can support third-party punishment, and these methods have shown promise experimentally in fostering cooperation (Köster et al., 2022). However, most techniques still rely on direct modifications of agent behavior through intrinsic rewards. These intrinsic reward depend on mechanisms such as, mimicking others' punishment behaviors (Vinitsky et al., 2023) and developing positive reputations (McKee et al., 2021). In contrast to these set of techniques, our method follows the key insight that human societies did not learn to be cooperative just through exploration and individual behaviour change. Rather, cooperation follows as a second-order effect once societies learn to coordinate their peer sanctions through social structures, such as informal norms and formal institutions (Richerson & Boyd, 2008; Henrich, 2016). Our work focuses on a particular manifestation of these structures, namely, classification institutions, that announce right and wrong behaviours around which agents can voluntarily coordinate their sanctioning behaviour (Hadfield & Weingast, 2012). More importantly, compared to previous work in MARL, we shift the focus from individual learning to learning about social structures. Specifically, our work uses standard MARL methods to give agents the

ability to recognize features of classification institutions (the *altars*) that represent the norms of a population.

**Norms and Institutions in multi-agent systems.** There is an extensive body of literature on norms in multiagent systems (MAS), with frameworks addressing various stages of a norm life cycle, including norm emergence, transmission, enforcement, and internalization within artificial agent societies (*c.f* Mahmoud et al. (2014); Chopra et al. (2018) for MAS literature and Gelfand et al. (2024) for an interdisciplinary review). In MAS, institutions typically represent norms using formal declarative languages, similar to logical specifications. For instance, the nADICO framework Frantz et al. (2013) provides a grammar for representing norms through institutional statements, and agents learn the content and enforcement of these norms by observing the behavior of others. In contrast to these symbolic norm representation methods, our approach uses a visual representation, eliminating the need for extensive handcrafted specifications in a formal language. Within the context of learning, another key difference lies in the focus of our work on learning enforcement behaviour rather than learning norm compliance Savarimuthu et al. (2024). In institutionalized MAS, norm enforcement through sanctions against violating agents is often a centralized process. Even when sanctions are imposed by third parties, the enforcer is typically a specially designed agent with dedicated monitoring roles Balke (2009); Balke & Villatoro (2011). In our approach, however, enforcement is entirely decentralized by making it part of the learned behaviour of each and every agent. In a related work, Garcia and Traulsen García & Traulsen (2019) analyze the effects of different pool punishment institutions, specifically pro-social and anti-social centralized institutions, where members can contribute to a coordinated punishment scheme. The model finds that public visibility of pro-social institutions is essential for the stability of cooperative strategies, as agents can condition their behavior based on the presence and visibility of these institutions. In comparison, our work do not deal with the question of establishment of the institution as part of the agent strategy, however, we analyze the impact of different types of institutions as well as visibility of institutions.

**Norm creation and emergence.** Embedding normative behavior in agents is commonly referred to as norm creation in the multiagent systems (MAS) literature Chopra et al. (2018). Previous approaches often treated this as an offline process, where agents were pre-programmed to follow specific norms, such as those related to property rights Conte & Castelfranchi (2006) or reputation (Hales, 2002). More recent approaches have introduced models of norm creation through specialized agents known as norm entrepreneurs Savarimuthu et al. (2007); Anavankot et al. (2024). However, norm creation through specialized agents raises additional questions, such as how and why norms are accepted and transmitted in a population Hoffmann (2005). This introduces the factor of network topology that determines agent interactions, and adds another layer of complexity to the process Sen & Sen (2009). Whereas norm creation is analyzed at a micro-level, norm emergence is studied as a macro phenomenon in artificial societies, often driven by a threshold effect—if a certain proportion of the population adheres to a behavior (descriptive norm) and enforces or expects the enforcement of that behaviour (social norm), that behavior can become widespread Morris-Martin et al. (2019). In this context, various models analyze the impact of independent variables at the micro-level, such as the cost of enforcement Savarimuthu et al. (2009), and environmental factors, such as network topology Zhang & Leezer (2009), on norm emergence.

## C TRAINING

In our experiments, we employed RLlib Liang et al. (2018) for training, utilizing the Proximal Policy Optimization (PPO) algorithm Schulman et al. (2017) to optimize agent behaviours. The agent architecture consisted of fully connected layers with hidden sizes of 64 and 256, using ReLU activations Agarap (2019). The environment was based on DeepMind's Melting Pot library Agapiou et al. (2023), with custom configurations designed to match the specific task objectives. We performed hyper-parameter tuning on network sizes and learning rate, train batch size and CNN filters using grid search on 200 configurations. Table 1 contains a list of parameters used during training. All experiments were run on a single GPU node with one A40 GPU and 32 CPUs. For this work, we consider a simple, practical and scalable approach to MARL is Independent Proximal Policy Optimization (IPPO), a decentralized method where each agent optimizes its policy independently using a variant of Proximal Policy Optimization (PPO). IPPO is well-suited for learning in multi-agent systems due to its ability to stabilize learning through clipped policy updates while maintaining scalability. Additionally, IPPO can be further enriched through inputs such as agent indication,

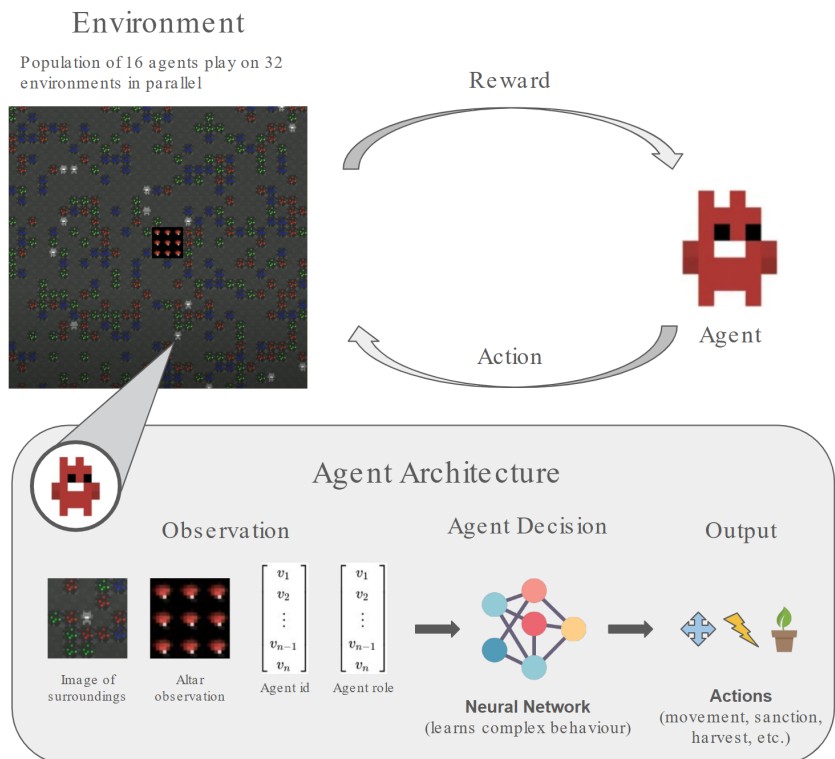

Figure 8: Agent architecture and interaction with environment.

which assigns unique identifiers to agents, and agent roles, which guide agents to adopt distinct strategies. These enhancements facilitate the learning of heterogeneous and independent policies, increasing the framework's flexibility and adaptability to diverse multi-agent dynamics.. These capabilities make IPPO the algorithm of choice for our investigation, allowing us to effectively study compliance and coordination in dynamic, multi-agent settings.

## C.1 AGENT ARCHITECTURE

The key objective of this work is to assess the implications of normative infrastructure on the alignment and cooperation capabilities of the agents. We are indeed proposing to shift the focus away from building complicated agent architectures in order embed norms and value in them and point the focus towards building normative infrastructure. Given this, we chose to conduct our experiments using a very simple shared-parameter architecture of our agents consisting of convolutions layers followed by fully connected networks. We tested both with and without GRU units and did not find significant difference in performance. Our architecture closely follows independent learning PPO de Witt et al. (2020b) agents, where the actor and the critic network are shared between agents. For heterogeneity in independently learning agents, we include both the agent indication and their pre-defined roles in the environment as extra input. We believe that our proposal is agnostic to the agent architecture and testing with more sophisticated agents is inteded as future work. Training details are available in Appendix C.

## D ADDITIONAL RESULTS

### D.1 ALLELOPATHIC HARVEST WITH FIXED ALTAR

Here, we consider an ablation of our Altared Allelopathic Harvest environment where the altar remains stationary throughout the episode (across all episodes) instead of being dynamic. The environment details remain the same as mentioned in Section E.1.1 apart from the changes discussed

| Parameter | Value |
|---|---|
| **Resources** | |
| Number of Rollout Workers | 30 |
| Number of GPUs | 1 |
| **Training** | |
| Seeds Used | 12345, 67890, 54321, 98765, 20242 |
| Rollout Fragment Length | 100 |
| Train Batch Size | 32,000 |
| SGD Minibatch Size | 4,096 |
| Number of SGD Iterations | 30 |
| Disable Observation Preprocessing | True |
| Use New RL Modules | False |
| Use New Learner API | False |
| Framework | torch |
| **Agent Model** | |
| Fully Connected Hidden Layers | (64, 64) |
| Post-FC Hidden Layer | (256) |
| CNN Activation | ReLU |
| FC Activation | ReLU |
| Post-FC Activation | ReLU |
| LSTM Use Previous Action | True |
| LSTM Use Previous Reward | False |
| LSTM Cell Size | 256 |
| **Experiment Trials** | |
| Stopping Criteria | 10,000 training iterations |
| Number of Checkpoints | 30 |
| Checkpoint Interval | 50 |

Table 1: Training and Hyper-parameter Configuration

in this section. We choose the altar to display red colored berry, making red monoculture the desirable outcome. We test three distinct conditions (similar to the versions presented in Section E.1.1) to explore the effects of different sanctioning mechanisms on agent's enforcement and compliance behavior and their ability to achieve a monoculture. We discuss these below:

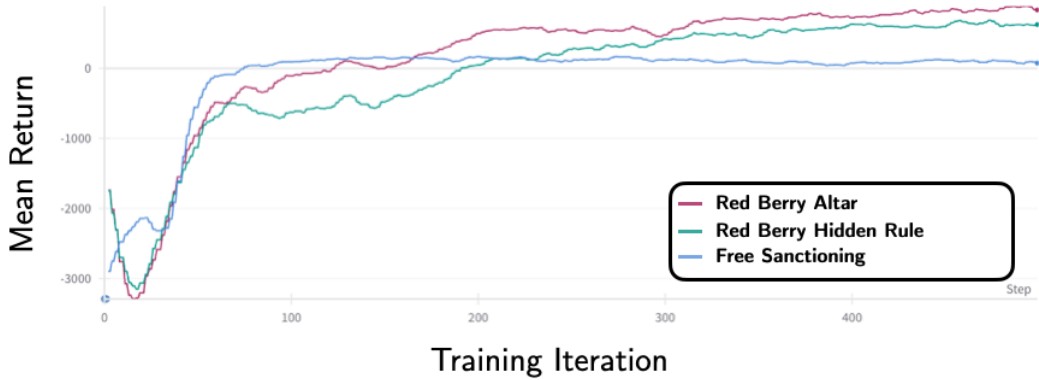

Figure 9: Rewards of training agents. Please note that we have adjusted the reward curves to only include reward for berry consumption and penalty for getting sanctioned while removing any effect of reward and cost received due to sanctioning rules

**Results**  Figure 9 shows agent performance in maximizing welfare, measured as the sum of rewards across agents and averaged over episodes. Welfare is maximized when agents align their planting and sanctioning behaviors to achieve a monoculture of one berry color, as faster ripening occurs with

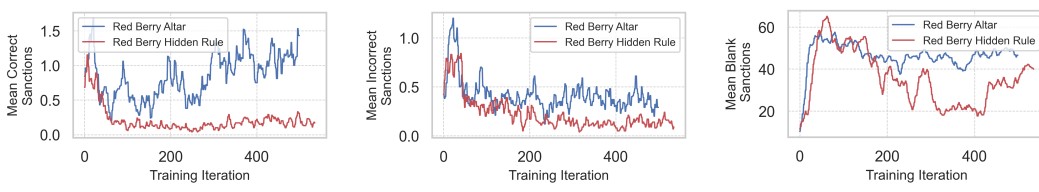

(a) Correct sanctions vs Training iterations

(b) Incorrect sanctions vs Training iterations

(c) Blank zaps vs Training iterations

Figure 10: Sanctioning Behavior of agents across training period

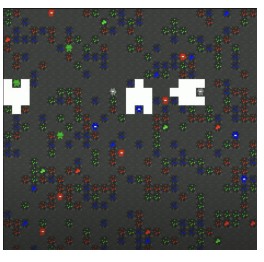
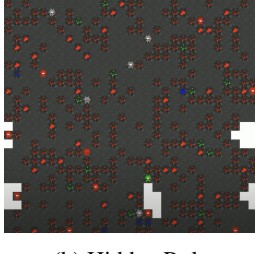
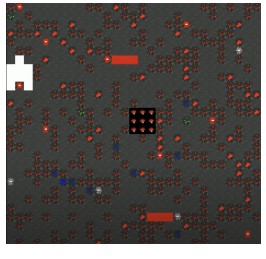

(a) Free Sanctions

(b) Hidden Rule

(c) Altared

Figure 11: Monoculture and Agents' status at halfway of an episode for trained agents

a larger fraction of the same color. Our results show that agents in the "altar" environment achieve the highest and most consistent rewards compared to the hidden rules baseline. We hypothesize that the altar observation helps agents correlate norm content with enforcement behavior, improving coordination and reducing training time. In contrast, agents struggle to learn compliance without signals in the free sanctioning condition.

To assess monoculture formation, we visualize the berry map midway through an episode (2000 steps) in Figure 13. In the free sanctioning condition, agents fail to align on a single berry color, limiting their ability to increase any fraction. However, in both the hidden rule and altar conditions, agents move toward monoculture, with over 95% monoculture achieved. Agents in the altar condition show better alignment on planting red berries, while the hidden rule condition has more free-riders planting other colors.

We also examine sanctioning behavior in Figure 10. Both the hidden rule and altar environments show reduced incorrect sanctioning over time, indicating that agents avoid wrongful punishments due to second-order punishment costs. Notably, agents in the altar environment increase correct sanctions, targeting free-riders who stop planting once monoculture is achieved. Random zapping in blank areas is frequent but not costly, as shown in Figure 10(c).

**Progress Towards Monoculture** Figure 13 highlights agents' progress toward monoculture throughout training. The top row (altar environment) consistently shows a higher monoculture fraction compared to the bottom row (hidden rule setup), aligning with the higher welfare observed. By iteration 400, most agents in the altar condition have stopped planting non-red berries, unlike the hidden rule setup. This demonstrates the altar environment's effectiveness in fostering coordination and increasing social welfare. Figure 12 shows the monoculture attained by a fully trained agent in Altated Allelopathic Harvest environment.

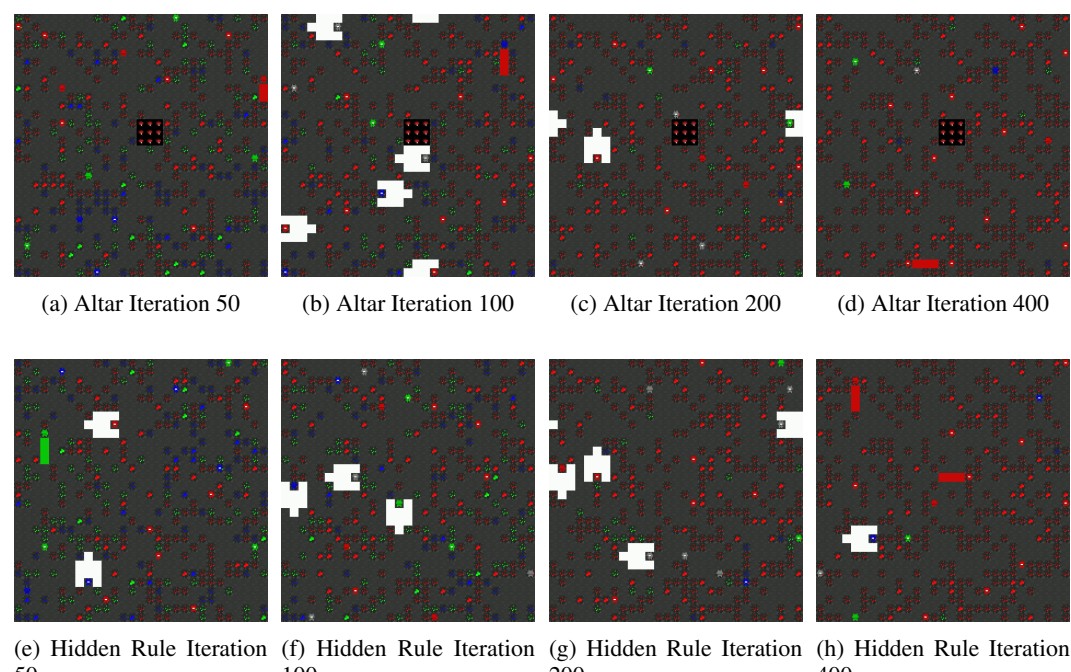

(a) Altar Iteration 50    (b) Altar Iteration 100    (c) Altar Iteration 200    (d) Altar Iteration 400

(e) Hidden Rule Iteration 50    (f) Hidden Rule Iteration 100    (g) Hidden Rule Iteration 200    (h) Hidden Rule Iteration 400

Figure 13: Monoculture and Agents' status at halfway of an episode for trained agents

# E    ENVIRONMENT DETAILS

## E.1    ALLELOPATHIC HARVEST

**Background and Setup**    The 'Allelopathic Harvest' environment (Agapiou et al., 2023; Köster et al., 2020) is a mixed-motive game which poses both the coordination and the free-rider problem, making it challenging for agents to reach a welfare maximizing outcome. It features a map containing a total of 348 berries of three different colors (116 of each red, blue and green) and sixteen agents that can plant and consume berries. Each agent has an intrinsic preference for a specific color berry. Out of 16 agents, 8 prefer red berries and other 8 prefer green berries by default. Agents get reward for consuming any ripened berry (+1) but receive higher reward for consuming their preferred color berry (+2). Agents can also plant berries of specific color but that does not generate any reward or cost and hence agents have no direct incentive to plant, leading to a free-rider problem. After planting a berry, agent's color changes to the color of the planted berry. However, after eating a ripened berry, their color is stochastically reset to gray The agents

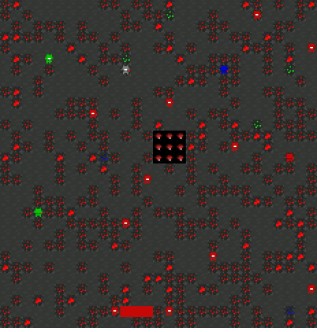

Figure 12: > 95% red monoculture.

can only consume ripened berries and the berry ripening rate is directly proportional to the fraction of the amount of berry of that color. Agents also have a zapping action which fires a white beam that they can use to tag other agents. When an agent is zapped (target), it receives a penalty of -10. While zapping, source agents don't receive any reward or penalty by default (but this may be changed in different versions of the environment below). An episode of this environment lasts 2000 timesteps. More details about it can be found in Agapiou et al. (2023).

### E.1.1    TRAINING ENVIRONMENTS

**Altar**    In this version, we introduce a **dynamic** 'altar' (Fig. 7) in the map – a visual observation ($3 \times 3$ subgrid) in the center of the map that displays berries of a specific color whose monoculture is desired. Agents have an augmented observation space that includes a memory slot, which starts as

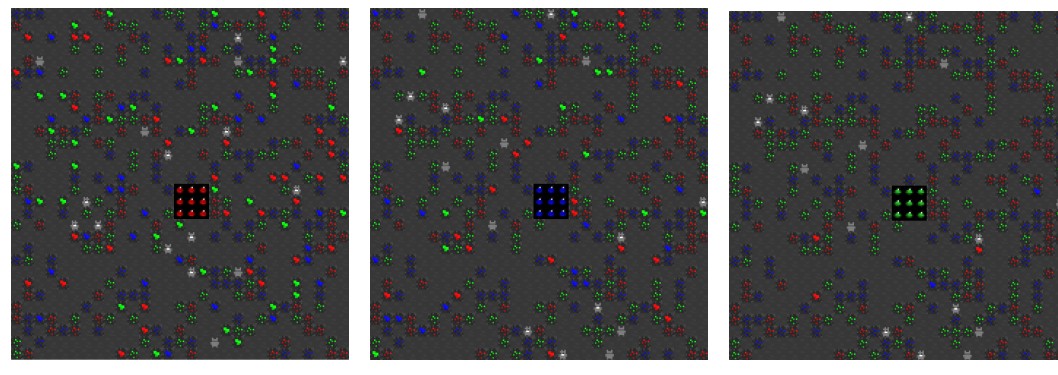

(a) Altar Prescribed Berry: Red    (b) Altar Prescribed Berry: Blue    (c) Altar Prescribed Berry: Green

Figure 14: Illustration of the `Altared Allelopathic Harvest` environment across different timesteps in an episode where the altar is present in the center of the environment depicting colored berries. The altar prescribed color changes periodically during an episode.

empty. When an agent enters a tile that is part of the altar, their memory slot updates to altar observation (Fig. 8). The altar is not stationary and its color changes dynamically. The altar color at the start of the episode is set to be randomly among red, blue and green. For the first 1000 timesteps, the altar changes color in a fixed manner at every 100 timesteps. However, after that, the color changes partially randomly in the following manner: `nextUpdateStep = previousUpdateStep + 160 + random(1, 100)` where `nextUpdateStep` denotes the timestep (in future) when the color needs to be changed. At each update, the next color is chosen such that it is *not the same as previous color*. More specifically, the next color is sampled randomly from the two remaining colors with equal probability for each.

Further, the presence of the altar also influences the reward dynamics associated with zapping (tagging). Specifically, if a source agent zaps a target agent of the same color displayed on the altar at that moment, both the source and target agents receive a penalty of -10 points. If the source agent zaps a target of any other color, the source agent receives a net reward of +10 points, while the target still receives a penalty of -10 points. Thus, the altar essentially only influences the reward (penalty) received by source (zapping) agent. There is no reward (penalty) if the zapping beam doesn't hit any agent.

We refer to this environment as `Altared Allelopathic Harvest` and provided an overview of it in Figure 14.

**Hidden Rules**    In the 'Hidden Rules' variant, the physical altar is removed from the environment, while all other dynamics remain the same as in the `Altared Allelopathic Harvest` version. Thus, the altar's influence persists only in the form of controlling tagging (zapping) related rewards and penalties without visual presence. If a source agent zaps another agent that is the same color as prescribed by the (hidden) altar, both the source and target agents receive a penalty of -10 points. If the source agent zaps an agent of any other color, the source agent receives a net reward of +10 points. We note that here also, even thought the altar is hidden, its color updates in the same manner as described in the `Altared` variant.

**Vanilla (Free Sanctioning)**    In this condition, there is no altar or hidden rule in the environment. Agents can freely zap other agents, with the target agent receiving a penalty of -10 points. The source agent, however, does not receive any reward or penalty for zapping.

### E.1.2    EVALUATION ENVIRONMENTS

In this section, we describe some extensions of the `Altared Allelopathic Harvest` environment which we used to evaluation of the agents trained in environments described earlier.

**Limited Altar**   In this variant, the altar vanishes from the environment and reappears periodically. At every `nextUpdateStep`, we alternate between either removing the altar from the environment (without changing its color so the agents still receive reward or penalty upon zapping other agents) or updating the color of the altar and reinstantiating it visually in the environment. For example, for the first 1000 timesteps, the altar disappears at $100, 300, 500, 700, 900$ step while it updates color and reappears at $200, 400, 600, 800, 1000$ step. When the altar is not present, the agents see blank (empty) cells on its place and don't receive any observation when stepping into it.

Thus, the altar is only visible and accessible for a limited number of steps. The objective of this environment is to test whether agents learn to optimize their visits to the altar to maximize updated knowledge within a restricted window. Fig. 15 provides an overview of this environment.

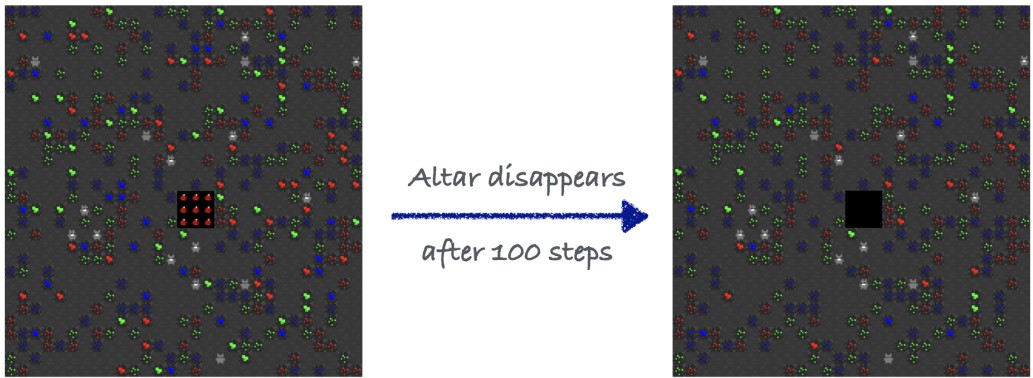

Figure 15: Illustration of 'Limited Altar' version of the `Altared Allelopathic Harvest` environment.

**Conflicting Altars**   In this environment, we introduce two more secondary 'altars' of size $2 \times 2$ and place them at the bottom-right and top-left regions of the map while the primary altar remains in the center. All three altars change their color periodically (at the same `nextUpdateStep`) but the secondary altars always show a color contradicting the primary (center) altar's prescribed color. However, the underlying reward dynamics associated with zapping only depends on the primary altar and changes when it updates. A key point here is when an agent visits a secondary altar, its observation gets updated wrongly.

The objective of this environment is to test whether the agents can figure out which altar (normative institution) is the "correct" one by visiting them and interpreting the signals received and then only choosing to visit the correct one at convergence. Fig. 16 provides an overview of this environment.

### E.2   COMMONS HARVEST

**Background**   The "Commons Harvest" environment is inspired by the 'Commons Harvest' substrate in Melting Pot (Agapiou et al., 2023), which itself draws from prior work on multi-agent reinforcement learning for common-pool resource appropriation (Perolat et al., 2017). In this environment, agents aim to collect apples scattered across six distinct patches. Each patch consists of multiple apple cells, with each cell having at least one neighboring apple.

Agents receive a reward of 1 for every apple consumed. Apples regenerate with a per-step probability that depends on the number of neighboring apples within an Euclidean distance of 2. Exact details about the regrowth probability can be found in Agapiou et al. (2023). Specifically, if there are no apples in the vicinity, the probability of regrowth is zero. Consequently, patches can be **permanently depleted** if all apples in a patch are harvested, requiring agents to exercise caution and avoid overharvesting. If agents exhaust a patch, it will not recover, and sustaining apple regeneration demands collective restraint among the agents. This dynamic leads to a *social dilemma*, akin to the tragedy of the commons, where individual incentives clash with the group's long-term interest.

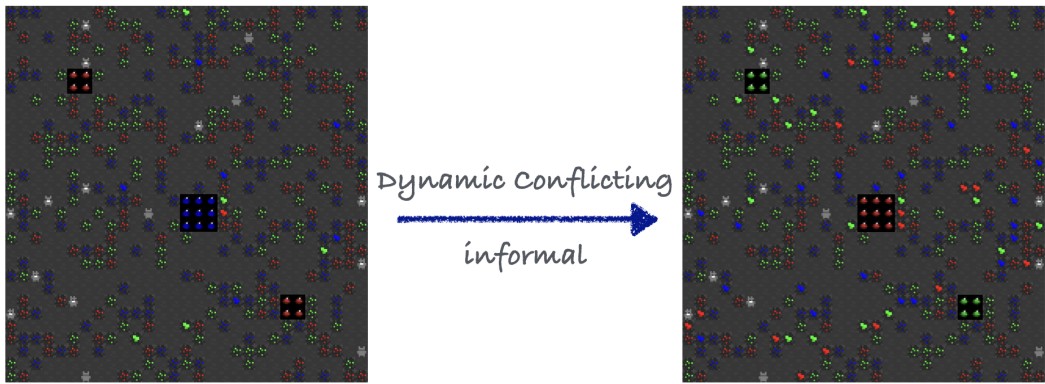

Figure 16: Dynamic Conflicting Institutions

Figure 17: Illustration of 'Conflicting Altars' environment where the secondary altars at the bottom right and top left region of the environment always show a color different from the primary altar (in the center) to distract the agents.

Agents face a strong incentive to consume the last remaining apple to maximize individual gain, potentially at the cost of losing that patch permanently.

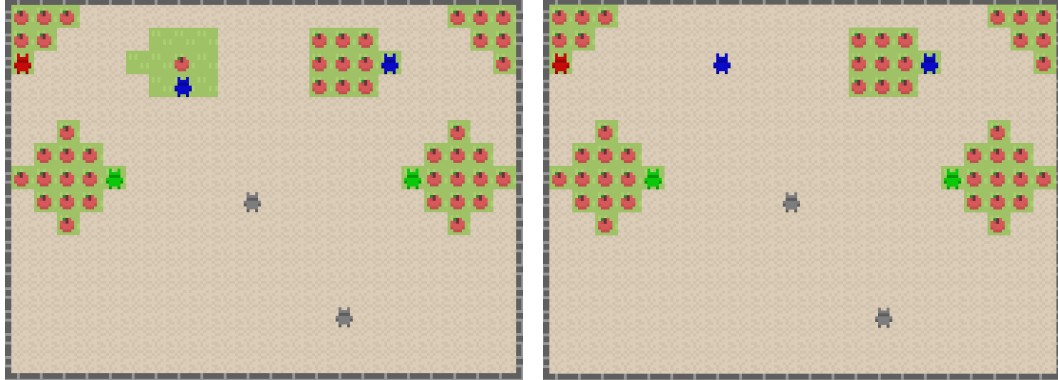

(a) Single apple left in one of the patches with a blue agent standing next to it.

(b) Blue agent eats the last remaining apple leading to the patch being lost permanently.

Figure 18: **Commons Harvest**: Illustration of how a patch can be lost irrevocably if the last remaining apple is eaten by any of the agents.

**Common Setup**    The environment features seven agents, six apple patches, and each episode runs for 5000 timesteps. Agents are initially colored gray. The six apple patches are grouped into three zones -— red (zone 1), green (zone 2), and blue (zone 3) -— with each zone containing two patches. When an agent eats an apple from a patch, its avatar color changes to the color of the corresponding zone, allowing agents to observe from which zone others have recently eaten.

As in the related "Allelopathic Harvest" environment, agents can also tag each other with a beam. If an agent is tagged, it is removed from the environment for 25 steps. No direct reward or punishment is received for tagging or being tagged, but there are indirect consequences: the tagged agent loses opportunities to collect apples during its timeout, while the tagging agent faces the opportunity cost of spending time on tagging rather than gathering apples.

**Altared Version**    In this setting, in addition to the existing setup, we introduce an 'altar' in the map – a visual indicator located at three positions: bottom left, bottom right, and center of the map. The altar consists of a 2x2 grid and displays the color of the zone from which agents should ideally

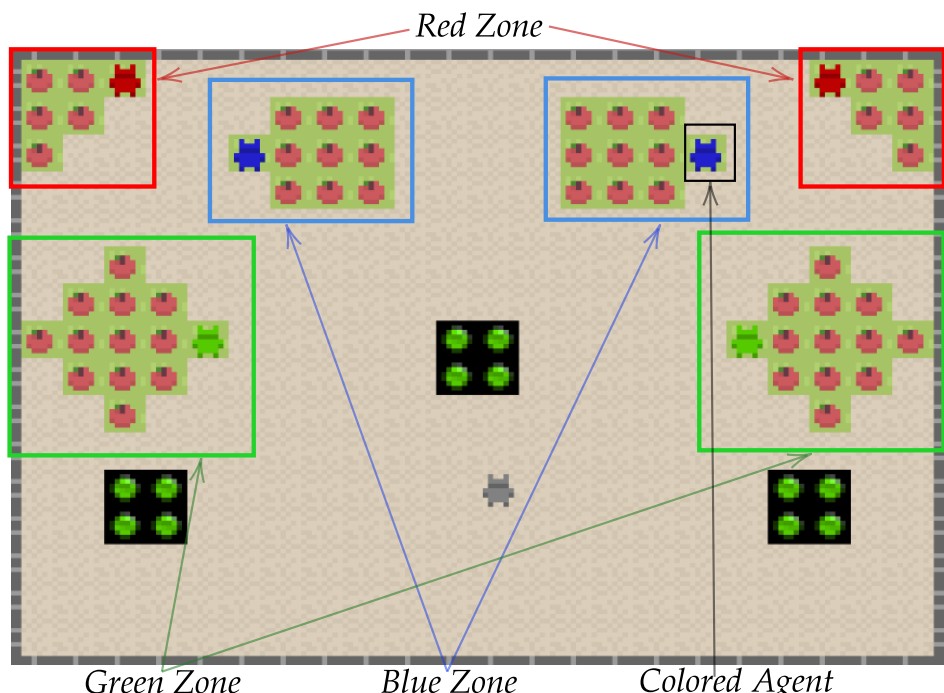

Figure 19: A snapshot from the 'Altared Version' of the Commons Harvest environment. Each colored box denotes an apple patch where the color indicates the zone to which the patch belongs. Agents obtain the color of the zone from which they last ate. Altar ($2 \times 2$ subgrid) is present in center, bottom left, and bottom right regions of the map. It is currently colored green indicating that apples should ideally be eaten from the green zone.

consume apples. When an agent enters the altar cells, it observes the altar color, which changes dynamically based on the number of apples remaining in each zone. A illustration of the different zones and altar is shown in Fig. 19 and the dynamics of the norms is illustrated in Figure 21.

The altar's color is initially set to the color of one of the zones whose patches' minimum apple count is maximum overall. It remains that color until one of the patches in the associated zone has less than 4 apples in which case its color is set to the zone whose patches' minimum apple count is maximum overall and above 3 at that moment. If no zone has both patches with more than 3 apples, the altar displays a yellow fire symbol, signaling that agents should refrain from consuming apples from any patch to avoid the risk of permanently losing them. The altar remains in the fire state until apple regeneration occurs.

When an agent tags another agent, the tagged agent is removed from the environment for 25 timesteps, similar to the original setup. However, the tagging (source) agent incurs a penalty of $-10$ if the tagged agent's color matches the altar color, indicating that the target agent was following the altar's guidance. Conversely, if the tagged (target) agent has a different color (excluding gray), the tagging agent receives a reward of $+10$. We show an illustration of the altar-based reward dynamics in Fig. 22a. When the altar is displaying fire, tagging any non-gray agent results in a $+10$ reward.

**Hidden Rules**  In the 'Hidden Rules' variant, the physical altar is removed from the environment, while the reward and penalty mechanisms remain the same as in "Altared Version". Thus, the altar's influence persists only in the form of controlling tagging (zapping) related rewards and penalties without actual presence. We show an illustration of this in Fig. 22b. Since agents no longer observe the altar directly, they would have to infer the altar's state based solely on the rewards they receive.

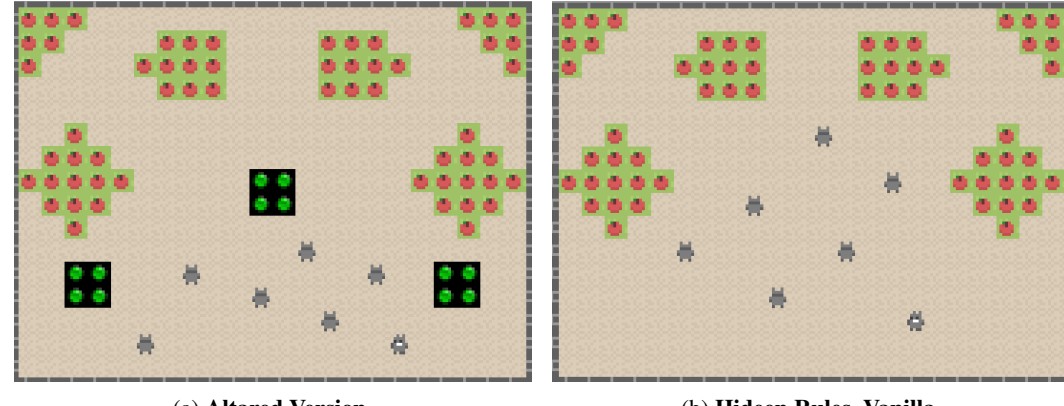

(a) **Altared Version**.
(b) **Hideen Rules, Vanilla**.

Figure 20: Snapshot of the initial frame in the different Commons Harvest environments.

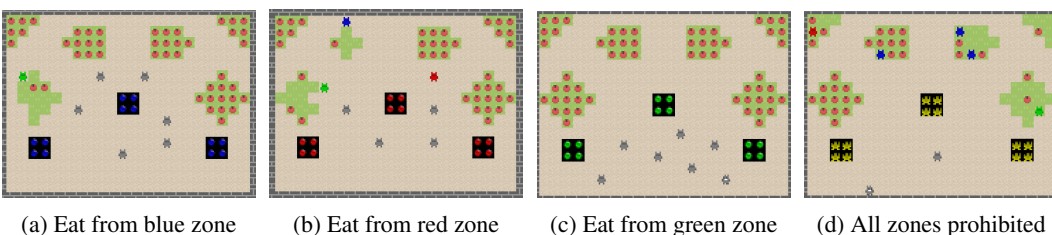

(a) Eat from blue zone
(b) Eat from red zone
(c) Eat from green zone
(d) All zones prohibited

Figure 21: Dynamics of ALTARED Commons Harvest

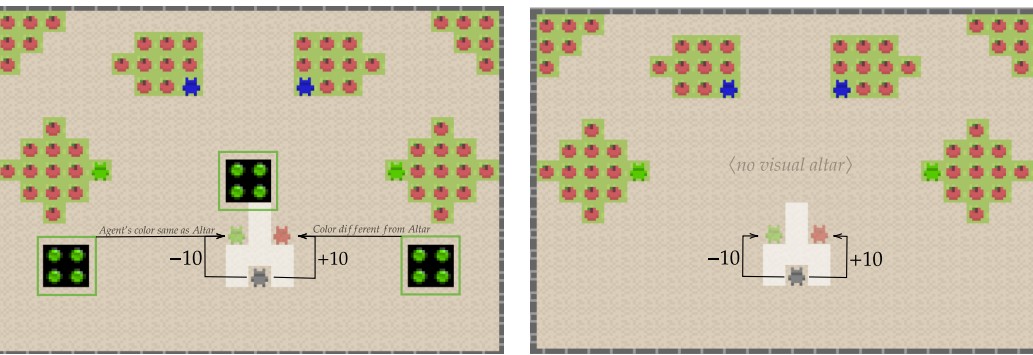

(a) **Altared Version**: The color of the altar is green and the grey agent fires a beam hitting both a green and red agent. The grey agent receives a penalty of $-10$ for zapping the green agent as its color is same as the altar but it receives a reward of $+10$ for zapping the red agent. Both the tagged agents are removed and reappear in the map after 25 timesteps.

(b) **Hideen Rules**: The altar is not present physically in the environment but its color is green. The grey agent fires a beam hitting both a green and red agent. It receives a penalty of $-10$ for zapping the green agent as its color is same as the virtual altar but it also receives a reward of $+10$ for zapping the red agent. Both the tagged agents are removed for 25 timesteps.

Figure 22: Illustration of the reward dynamics based on the altar in different versions of the Common Harvest environment.

**Vanilla**    In the 'Vanilla' variant, the altar is not present, and the rewards and penalties associated with the altar are also eliminated. The agents receive rewards only for consuming apples, and receive no other signal from the environment. A snapshot of this environment at the beginning is shown in Fig. 20b.

