# OpenReview forum: "Altared Environments: The Role of Normative Infrastructure in AI Alignment"
_ICLR.cc/2025/Conference — Submitted to ICLR 2025_

### Official Review · Reviewer_DY2Z · 2024-10-26

**Soundness:** 2
**Presentation:** 1
**Contribution:** 2
**Rating:** 3
**Confidence:** 4

**Summary:**

This paper incorporates 'altars', institutions within selected cooperative multi-agent environments by DeepMind (Melting Pot), that provide agents with observable guidelines on how to punish each other's actions to enhance collective welfare. Authors claim to validate their hypothesis by running 3 experiments, 1. "vanilla" trained using PPO 2. hidden-rule based and 3. altars as environment features

**Strengths:**

1. Well-written paper, reads naturally (flow, language)
2. Introduction and Preliminaries set the stage very well and explain the broader interest in the problems touched.

**Weaknesses:**

## Main concern:
1. Experiments and their results do not support the claims made by unfair comparison. For example, we compare mean rewards while one of the baselines has a clear disadvantage in earning rewards.
2. Code is not accessible to reproduce or investigate the work further
3. No description on how the altar "dynamic" attribute works
4. No mention of the results of the original authors of the Melting Pot environment suite

## Sub-issues:
1. The abstract is very ambiguous, assuming specific knowledge of the environments, which can confuse readers who are not knowledgeable.
2. I appreciate the anonymization of the repository, however it renders empty, this leads to missing code, to reproduce the work and experiment videos cannot be accessed.
3. Minor spelling mistakes such as in line 128, 515
4. Line 320 - link to reference is missing
5. Figure 1 is an important figure however, it is not / hard to understand
6. Figure captions are insufficient for most / all figures in the main body of the paper.

**Questions:**

## Questions:
1. It is not clear to the reader how the altar is set up, i.e. How are punishment policies chosen? Is this hard-coded? This is what is being revealed: "Finally, as the institutional content is dynamic," but there is no indication of how this works.
2. The same goes for the hidden-rules setup. How are hidden rules decided, and what are they exactly? This might become clear when investigating the environment suite; however, it should be made clear in the main body of this paper.
3. The baselines are not supporting. The results reported by the original authors of the Melting Pot environment suite are not mentioned, and I believe this work should be compared against what the original work reported.
4. The experiment results show how the altar and/or hidden-rule-based setups surpass the "vanilla" setup when looking at Mean Episode Reward. However, the chosen environment scenarios are not about mean episode rewards. What they care about is, for example, the depletion rate.
5. Your Vanilla baseline mentions that zapping results only in negative rewards, no positive rewards, while the other setups yield positive rewards. That means we do not have an apples-for-apples comparison of results.
6. Finally, my biggest question is: What if the agent's reward function is modified, such as the altars policies are set? Adding the altar as an observable sanctioning feature appears to be a complicated dynamic reward that is not given by the environment directly but rather potentially by other agents catching others doing the wrong thing. This would mean that if a negative reward for bad behaviour is always given, we should even see better performance, but this could be achieved by reward-shaping or adding a reward structure already only. In conclusion, the partially observable rules that help agents police each other's actions, when observed, seem not worth the effort.

---

> ### Author Response · Authors · 2024-11-29
> **Response to Reviewer DY2Z (1/3)**
>
> We thank the reviewer for their thoughtful feedback and for recognizing the paper’s natural flow and clarity. We are especially pleased that the Introduction and Preliminaries effectively frame the broader interest of the work. We have conducted a careful revision to address the concerns raised, which we discuss below.
>
> > my biggest question is: What if the agent's reward function is modified, such as the altars policies are set? Adding the altar as an observable sanctioning feature appears to be a complicated dynamic reward that is not given by the environment directly but rather potentially by other agents catching others doing the wrong thing. This would mean that if a negative reward for bad behaviour is always given, we should even see better performance, but this could be achieved by reward-shaping or adding a reward structure already only. In conclusion, the partially observable rules that help agents police each other's actions, when observed, seem not worth the effort.
>
> We are not entirely sure we fully understand the reviewer’s question, so we will first outline our interpretation and then respond accordingly. Based on the comment, we interpret that the reviewer is suggesting replacing the altar and its associated enforcement mechanism with existing approaches, such as reward shaping. Specifically, this would involve directly modifying the agents’ reward functions to penalize bad behavior, thereby eliminating the need for the altar as an explicit mechanism for agents to enforce norms. The reviewer seems to suggest that using the altar introduces unnecessary complexity and is not worth the effort. If this interpretation is incorrect, we welcome the reviewer to clarify their comment further.
>
> **Response:** Based on this interpretation, the approach proposed by the reviewer is in direct contrast to the objectives of this work. As detailed in the introduction (and clarified further in the revised version), this work aims to shift the focus of AI alignment research from embedding norms and values into AI agents to building normative infrastructure that supports agents in achieving alignment. Approaches like reward shaping (whether using intrinsic or extrinsic rewards) impose a significant cognitive and exploratory burden on the agents themselves, which this work aims to avoid. As we argue in the introduction, humans do not internalize all norms and values; instead, they rely on cooperative interactions with institutions to achieve alignment in groups. This forms the core motivation for our framework and experimental study: investigating the role of institutions in helping AI agents adapt with dynamic norms and stabilize around cooperative outcomes in comparison to environments without institutions.
>
> Regarding the reviewer’s reference to third-party enforcement or “agents catching others doing the wrong thing,” this concept has a long history of study in social anthropology, economics, and law. As noted in the introduction and recognized by Reviewer HNPm, such mechanisms are foundational in human societies and are widely observed in real-world social systems. Our approach builds on this conceptual framework and is grounded in the Hadfield-Weingast model [1], which formalizes the interplay between classification institutions and enforcement mechanisms.
>
> On the comment that the altar introduces a “complicated dynamic reward,” we believe the reviewer is conflating the relation between the altar role and the reward mechanism itself. This is likely due to a lack of clarity in the earlier version, which we apologize for and have addressed in the revision. The key clarification is that the sanctioning reward mechanism remains identical between our baseline (Hidden-Rule SMG) and our approach (Altared SMG). In both cases, norms are dynamic, but in our approach, they are explicitly observable through the institution. The principles governing sanctioning are as follows:
> - Sanctioning is costly for both the source (agent who sanctions) and the target (agent being sanctioned).
> - Sanctioning becomes rewarding for the source only if the target had violated the prevailing norm in the previous timestep.
>
> In the baseline setup, agents must infer norms and adapt to changing norms solely from the sanctioning reward signal. In our approach, the agents also receive an explicit signal for the prevailing norm via the altar, requiring them to learn to correlate altar observations with the sanctioning reward mechanics. Our results show that agents in the altar setup consistently visit the altar, engage in correct sanctioning behavior more often, and achieve better alignment compared to the baseline.
>
> [1] What is law? A coordination model of the characteristics of legal order, Gillian K Hadfield and Barry R Weingast, Journal of Legal Analysis, 2012.
>
> > Missing Citations and Captions
>
> We have addressed the missing citations and revised the figure captions throughout the paper to ensure they are more informative and descriptive.

---

> > ### Comment · Reviewer_DY2Z · 2024-11-29
> >
> > Thank you for your detailed answer to my questions and concerns.
> >
> > A quick summary of what I understand from the paper:
> > 1. Agents go to the altar to check on the latest sanctioning norms (infer norms).
> > 2. If agents observe other agents doing something sanctionable, according to the altar, they can sanction them and get a positive reward for correctly sanctioning.
> > 3. If agents sanction others incorrectly, they get a negative reward.
> >
> > If this is correct, these are my concerns:
> > - I don't understand why we still would not be interested in metrics/objectives that are part of the original environment. The sanctioning mechanism aims to increase performance on such metrics / objectives.
> > - With the results and experiments, you are showing that your mechanism helps agents align on dynamic norms, but what for?
> > - There are no fine-grained results / detailed results, that means potentially the only thing agents get right is sanctioning while they completely fail to complete the tasks / objectives part of the environments.
> > - What the results show, however, is that altar visits decline in both environments while correct sanctioning declines and incorrect sanctioning inclines and as a result, the Adjusted Mean Episode Reward still increases. This means that agents have learned not to go to the altar as much; however, they get worse at sanctioning. On the other hand, the Adjusted Mean Episode Reward still increases. This makes no sense to me. What is contributing to the Adjusted Mean Episode Reward if the correct sanctioning is declining? It cannot come from correctly sanctioning. In order to answer this question, we are missing results.

---

> ### Author Response · Authors · 2024-11-29
> **Response to Reviewer DY2Z (2/3)**
>
> > Experiments and Vanilla Baseline
>
> We appreciate the reviewer’s concerns and believe there may have been a misunderstanding regarding the reporting of results, particularly for the vanilla baseline. The vanilla baseline is designed to represent the original environment as closely as possible, retaining its original sanctioning mechanism. In the original setup, sanctioning resulted in a penalty for the target agent but provided no reward or penalty for the sanctioning agent. This is retained in the vanilla baseline. By contrast, both the Hidden Rule SMG and Altared SMG setups include a reward structure for correct sanctioning to incentivize agents to learn sanctioning behavior. The altar-based approach further adds explicit norm representation as a feature of the environment, as discussed earlier.
>
> **Regarding fair comparison**, we emphasize that we report Adjusted Mean Reward for both the Hidden Rule SMG and Altared SMG setups. This adjustment excludes the artificial positive rewards gained from correct sanctioning, which are designed to train agents, and instead includes:
> - Base environmental rewards (e.g., consuming a berry or harvesting an apple).
> - The cost of sanctioning.
> - The cost of being sanctioned.
>
> This adjustment ensures that the reported rewards reflect meaningful cooperative performance by focusing on sustained outcomes. It demonstrates that the sanctioning mechanism is beneficial only when the environmental rewards outweigh the costs incurred through sanctioning, providing a fair and transparent comparison.This presents even stronger results, as both the Hidden Rule SMG and Altared SMG setups perform better than the vanilla baseline, even when including the cost of sanctioning for the source agent—a feature absent in the vanilla baseline.
>
> **Regarding metrics**, we agree that metrics such as depletion rate are important in these environments. Sustained high welfare inherently requires agents to coordinate toward socially optimal strategies. For example: in Commons Harvest, agents must avoid overharvesting to prevent depletion, as persistent overharvesting directly reduces overall rewards in subsequent episodes.
> In Allelopathic Harvest, agents must learn to align with shifting norms to maintain optimal resource use. While Adjusted Mean Reward reflects this coordination, we also report additional metrics directly relevant to norm enforcement, such as rates of correct and incorrect sanctioning. Finally, as shown in Figure 4c, our approach indeed significantly reduces the depletion rate over time. These results indicate that agents not only comply with evolving norms but also enforce them effectively in the presence of institutions.
>
> We hope this clarification addresses the reviewer’s concerns. We welcome further feedback to ensure the metrics and comparisons are clear and meaningful.
>
> > No mention of the results of the original authors of the Melting Pot environment suite
> The baselines are not supporting. The results reported by the original authors of the Melting Pot environment suite are not mentioned, and I believe this work should be compared against what the original work reported.
>
> We respectfully disagree with the reviewer and believe that a comparison with the original results from the Melting Pot environment suite would not be meaningful for the following reasons:
>
> **Different Focus and Objectives:**
> Our work does not propose a novel AI algorithm to solve the alignment problem. Instead, we focus on environments that pose normative alignment challenges for groups of AI agents and propose a formal framework for augmenting such environments with classification institutions. The goal is to systematically investigate the impact of explicit institutional mechanisms on agents’ behavior, specifically their ability to engage in enforcement mechanisms and work toward alignment. Thus, our comparisons are between environments (with and without institutions) rather than algorithms. As we also clarify in response to another question and in our revised introduction, our study deliberately uses architecturally simple agents to minimize their cognitive and exploratory burden, making comparisons with complex algorithms, such as those tested in the original Melting Pot work, not applicable.
>
> **Modifications to Melting Pot Environments:**
> While we utilize the Melting Pot suite for its high-fidelity implementation of social simulations, we modify the environments to suit the objectives of our study, as detailed in the paper. These modifications ensure the environments are tailored to our investigation of dynamic norms and institutional mechanisms.
>
> > Missing Code and Ambiguous Abstract
>
> We apologize for the oversight regarding code accessibility. The code has now been made available at the same link, enabling further investigation of our work.
>
> We have rewritten the abstract to ensure it is precise, self-contained, and accessible to readers unfamiliar with the specific environments.

---

> ### Author Response · Authors · 2024-11-29
> **Response to Reviewer DY2Z (3/3)**
>
> > No description on how the altar "dynamic" attribute works.
> > It is not clear to the reader how the altar is set up, i.e. How are punishment policies chosen? Is this hard-coded? This is what is being revealed: "Finally, as the institutional content is dynamic," but there is no indication of how this works.
> > The same goes for the hidden-rules setup. How are hidden rules decided, and what are they exactly? This might become clear when investigating the environment suite; however, it should be made clear in the main body of this paper.
>
> We apologize for the unclear writing in the initial submission that discussed all these questions but in less readable manner. To remedy that, we have not revised the manuscript comprehensively to make all these details very clear, precise and accessible. We first outline the specific revisions to the sections relevant to the concerns raised by the reviewer and then briefly describe them here.
> - In **Section 2**, we introduce the theoretical framework that underpins our methods, provide the formalism of the Sanction Augmented Markov Game (SMG), and discuss how this framework serves as the foundation for the Hidden Rule SMG baseline used in our experiments.
> - In **Section 3**, we address the key challenge of **dynamic norms** in our setup, offering a formal discussion of this issue. We then present one of our main contributions: the extension of Markov Games to their altered versions, referred to as Altared Games.
> - The exact nature and dynamics of the norms for each environment, i.e. how we instantiate the rule sets and how they change over time are discussed in detail in **Appendices A.1** and **A.2** of the revised version.
>
> **Dynamic Norms:** For both the hidden rule setup (now called Hidden Rule SMG) and altar setup (now called Altared SMG), the dynamic norm mechanism works in the same way. In fact, all the setup between Hidden Rule SMG and Altared SMG is exactly the same except the presence of institutional features of the environment, called altar  in our approach.The controlled nature of our framework ensures that external factors do not interfere with the experimental results, making it possible to isolate and analyze the specific impact of the altar. By holding all other aspects of the system constant, we enable hypothesis testing that focuses solely on the presence or absence of an explicit classification institution. This methodology allows for a rigorous examination of how agents’ learning, coordination, and compliance behaviors change when norms are made observable
> In our formal setup, the evolution of norms is governed by an update function: $Nt+1 = f (Nt, \phi)$, where $f$ captures the mechanism of norm evolution, and $\phi$ represents triggers or drivers of change. We do not model the determinants of norm evolution but these drivers could be thought of as arising from external inputs (e.g., regulatory updates or environmental changes), agent-driven mechanisms (e.g., collective decision-making or voting), or stochastic events (e.g., resource depletion or unexpected disturbances).
>
> In this work, the norms and corresponding punishment policies are hard code i,e,wWe assume that a normative system is already in place, with predefined but dynamic norms (as discussed in Section 3) and associated enforcement mechanisms. Our focus is on investigating the role of explicit institutional mechanisms, such as the altar, in improving agents' ability to learn and adapt to these evolving norms, particularly in dynamic environments. To the best of our knowledge, the role of institutions in supporting norm compliance for AI agents has not been studied before. Hence, we focus on demonstrating that an ideal normative system produces higher payoffs when agents have access to institutions that enhance their capacity to learn and track the content of norms. This allows agents to adapt more rapidly and reliably to a changing normative structure, irrespective of how those norms originated. While broader questions regarding the emergence and establishment of norms are beyond the scope of this work, readers can refer to relevant literature, such as [1,2] for a detailed discussion.
>
> We will include this discussion in the final version of the paper.
>
> [1] A survey of research on the emergence of norms in multi-agent systems, Bastin Ton Roy Savarimuthu and Stephen Cranefiled, 2011.
>
> [2] Norm emergence in multiagent systems: a viewpoint paper, Andreasa Morris-Martin, Marina De Vos and Julain Padget, AAMAS 2019.
>
> We believe that the updates and responses we have provided address the key concerns raised. We kindly invite the reviewer to revisit their score in light of these clarifications and improvements. If there are any remaining questions or areas where further discussion would be helpful, we would be delighted to engage further and ensure all concerns are fully addressed.

---

> ### Author Response · Authors · 2024-11-30
> **Response to Reviewer DY2Z (1/3)**
>
> We thank the reviewer for their prompt response for summarizing their understanding of the paper. We’re afraid that the reviewer's comments display several misunderstandings  of our approach and setup, which seems to have been exacerbated even further by misreading of our overall results and plots. To remedy that, we will first describe our approach briefly here and then respond to specific comments about our results.
>
> > A quick summary of what I understand from the paper:
> >- Agents go to the altar to check on the latest sanctioning norms (infer norms).
> >- If agents observe other agents doing something sanctionable, according to the altar, they can sanction them and get a positive reward for correctly sanctioning.
> >- If agents sanction others incorrectly, they get a negative reward.
>
> While the above summary is correct, it is incomplete and so may be the source of misunderstanding. Agents do learn to visit the altar (that’s an important and interesting result in and of itself) and they do get positive rewards for correct sanctioning and negative if incorrectly. But the reason that correct sanctioning produces group payoffs is that it aligns the entire group on a better collective behaviors: all agents plant the same color berry in Allelopathy and all agents avoid harvesting from depleted apple patches in Common Harvest. Because the entire group is coordinated on socially-beneficial behaviors, individuals can consume more and this is the main source of reward in the environment. The average reward metric is primarily based on the level of consumption agents achieved, adjusted for the costs incurred from sanctioning and being sanctioned. We do not include the rewards for correct sanctioning as sanctioning is group-beneficial only to the extent that it generates higher consumption by helping to solve the social dilemma of over-harvesting or uncoordinated planting. Note also that if sanctioning is reliable, agents learn to predict this and avoid it: by complying with the current norm.  In a successful group, there will be high rates of compliance and hence low rates of actual sanctioning.
>
> This comment helped us to recognize that our use of the term “rewards” in our principal metric of Adjusted Mean Rewards was confusing. Although we clearly defined the metric on line 462 in the paper and so its meaning can be seen in the math, we have changed the name of the metric to **Collective Return**, the term used in the original papers (and in Koster et al 2022, on which we build). This focuses on the core behaviors of interest from a welfare point of view: benefits for agents generated from consumption and costs associated with sanctioning. Again, rewards for correct sanctioning do not enter into the Collective Return (formerly Adjusted Mean Episode Rewards) metric.
>
> > I don't understand why we still would not be interested in metrics/objectives that are part of the original environment. The sanctioning mechanism aims to increase performance on such metrics / objectives.
>
> We ARE interested, and report on, metrics and objectives from the original environment from Perolat et al (2017) and Koster et al (2020). Note that agents in both papers also have sanctioning mechanisms (zapping beam); we are not testing the impact of sanctioning per se on these metrics. The most important metrics are those that measure how successful the agents are at solving the social dilemmas: increasing the apples or berries they can consume. This is what our Adjusted Mean Episode Reward Metric (now renamed Collective Return, as we mentioned above) captures in both environments. Perolat calls this “Efficiency”, Koster (2022) calls this “Collective Return”. We also report apple depletion (a metric in Perolat) and are still working to report monoculture (this is much harder to measure in our dynamic setting; our appendix shows a monoculture metric for the static case which is more directly comparable to Koster 2020.) The measures in our environment and theirs are not directly comparable.
>
> Notably, their sanctioning mechanisms are costless to use–ours are costly. Ours are incentivized both by private benefits (as in Perolat and Koster) and by social benefits (rewards for correct sanctioning). Most importantly, however, we want to emphasize that the goal of our work is not to develop an algorithm that outperforms those in these original papers; so we are not measuring our performance against those benchmarks.  We are testing the hypothesis, grounded in Hadfield & Weingast (2012), that an institutional feature improves performance in the sense of improving Collective Return. That test is conducted by comparing our Hidden Rule metrics to our Altar metrics. Our results support the hypothesis and generate the important result that improving performance on a social dilemma for multiagent systems can rest not ONLY in improving algorithms, but also in improving environments.

---

> ### Author Response · Authors · 2024-11-30
> **Response to Reviewer DY2Z (2/3)**
>
> > With the results and experiments, you are showing that your mechanism helps agents align on dynamic norms, but what for?
>
> We admit to being very confused by this question as it suggests the reviewer has not understood the most basic goals of our paper. If agents align on dynamic norms, the ‘what for’ is: to better solve the fundamental social dilemma posed in the substrate (Commons Harvest or Allelopathy). We selected the norms to be ones that improve Collective Return by reducing depletion of a limited resource or by coordinating agents on a monoculture and limiting free-riding. It of course follows directly that if we can show improved norm-following (because of improved correct sanctioning in accordance with the norms), then we show improved performance (which we measure.) Our hypothesis is that norm-following is improved with an altar because the altar improves sanctioning when norms are changing (because the best solution to the social dilemma changes) and this is exactly what our results show. That’s what all of this is “for”.
>
> > There are no fine-grained results / detailed results, that means potentially the only thing agents get right is sanctioning while they completely fail to complete the tasks / objectives part of the environment.
>
> Again, this comment is very worrisome to us and we hope it is only because of the core misunderstanding about what is measured by what we called Adjusted Mean Episode Reward in the paper and now call Collective Return. It is absolutely not the case that all our agents get right is sanctioning–what they get right because of correct sanctioning is planting or consumption behavior. They plant more and plant the correct color more successfully in Allelopathy and they avoid consuming apples from patches at critical risk of depletion in Commons Harvest. We present the fine-grained results about Collective Return in both cases which show the impact of these behaviors. We report depletion in Commons Harvest. We admittedly do not show the monoculture fraction in the main paper–as this is very hard to measure in the dynamic setting and we are still working on it–but we report this metric as a ‘sanity check’ in the Appendix D for the static norm case. We will also include videos that show the detailed behaviors in the final version. We’re not sure where this comment is coming from, other than a misunderstanding.

---

> ### Author Response · Authors · 2024-11-30
> **Response to Reviewer DY2Z (33/3)**
>
> > What the results show, however, is that altar visits decline in both environments while correct sanctioning declines and incorrect sanctioning inclines and as a result, the Adjusted Mean Episode Reward still increases. This means that agents have learned not to go to the altar as much; however, they get worse at sanctioning. On the other hand, the Adjusted Mean Episode Reward still increases. This makes no sense to me. What is contributing to the Adjusted Mean Episode Reward if the correct sanctioning is declining? It cannot come from correctly sanctioning. In order to answer this question, we are missing results.
>
> The concerns here again stem from a misunderstanding about our core metric, Collective Return (originally called Adjusted Mean Episode Reward). What contributes to Collective Return is higher consumption. These results are not missing; they are the core metric we report.
>
> The reviewer also has misunderstood the altar visit metric and they have misread the sanctioning plots. Let us clarify the following:
>
> - Altar visits are reported in **Figure** 4 for both the environments. We request the reviewers to check the Y-axis . Initially agents are visiting the altar at a significantly high rate. It is expected that this rate will drop significantly as the agents learn the environment. Visiting an altar has an opportunity cost – during this time, agents could have instead consumed berries or apples.
> As discussed in the details in **Appendix A.1** and **A.2**, the norms do not change very frequently, hence it does not make sense for agents to keep visiting the altar and over time. They stabilize around number of visits which is around 30 for Allelopathic harvest and around 500 in Commons Harvest (the high number in Commons Harvest is because there are three altars and smaller map so easier to visit them repeatedly, less opportunity cost).
> - Correct Sanctioning for our approach (Altared SMG) **always increases and improves** over the baseline Hidden Rule SMG (see Figure 2b) for Commons Harvest environment. Incorrect sanctioning for our approach (Altared SMG) **always decreases and stays below**  the baseline Hidden Rule SMG (see Figure 2c).
> - For Allelopathic Harvest, Correct Sanctioning slightly decreases for Altared SMG and incorrect sanctioning slightly increases in the center of training. This is plausible because free-riding (grey agents) is considered a violation of norm throughout the episode and hence initially there are many free-riding agents, and sanctioning them is a correct action. But as soon as agents learn not to freeride, the number of correct sanctions coming from that source declines. In any case, we again refer the reviewer to look at the y-axis of **Figure 3b** and **Figure 3c**. The overall correct sanctions still remain above 60%  and the sanctioning behavior of Altared framework is improved over the Hidden Rule SMG towards the end.
> - Again, to emphasize, Collective Return (formerly Adjusted Mean Reward) should not be increasing or high just because of correct sanctioning, as sanctioning reward is not included in that number. It is indeed high because agents are learning to improve on the base environment rewards.
>
> We hope these responses help to allay the reviewer’s concerns.We  welcome the reviewer to raise further clarification questions if there still remains any confusion and  we would be happy to help clarify them.

---

> > ### Comment · Reviewer_DY2Z · 2024-11-30
> >
> > To summarize:
> >
> > - In your initial comment, you mentioned that the original objectives were not relevant to your work and even argued that it would not make sense to refer to the results of the original authors of the environment as a baseline.
> > - In your most recent comment, however, you appear to express the opposite, indicating that you "ARE" indeed interested in the original metrics.
> > - Finally, fine-grained results for your method have not been provided.
> >
> > Based on this, I feel confident in my decision to maintain the scores I have given. I want to sincerely thank you for your efforts and for the detailed explanations you have provided.
> >
> > I kindly suggest clarifying all potential sources of misunderstanding in the manuscript and ensuring that all fine-grained results for your experiments are provided. This should include all metrics relevant to the environment as well as those specifically pertinent to your mechanism.

---

> > > ### Author Response · Authors · 2024-12-01
> > > **Further response to DY2Z**
> > >
> > > The reviewer's response does not reflect adequate engagement with our detailed effort to address their original review and questions, all of which we answered effectively, especially by noting that the reviewer had made a basic mistake in interpreting our core metric and our research question.
> > >
> > > In the initial review, the reviewer mentions their biggest question to be:
> > > > What if the agent's reward function is modified, such as the altars policies are set? ...
> > >
> > > We provided a detailed answer to that question but the reviewer never acknowledged it.
> > >
> > > In the follow-up questions, the reviewer stated another big issue:
> > >
> > > >This means that agents have learned not to go to the altar as much; however, they get worse at sanctioning. On the other hand, the Adjusted Mean Episode Reward still increases. This makes no sense to me. What is contributing to the Adjusted Mean Episode Reward if the correct sanctioning is declining?
> > >
> > > What is the reviewer's response now that we have clearly answered this question?
> > >
> > >
> > > Further we find the reviewer's current response to be argumentative and cursory. The reviewer appears not to understand the difference between saying we are interested in a set of metrics, which we use, and saying that it is inappropriate to compare results directly across papers as a ‘baseline’ (which we said both in our original draft and in our latest response). We hope the reviewer really did not mean we should compare to the “baseline” in the “original” Melting Pot paper as those are for a completely different setup. The reviewer fails to explain why they think differently.
> > >
> > > Finally, the reviewer keeps asking for “fine-grained results” but has nowhere said what they mean by that. We have no idea what the reviewer means.

---

> ### Comment · Reviewer_DY2Z · 2024-12-01
>
> Thank you for your feedback.
>
> By fine-grained results I'm asking for the bare minimum: Metrics that support your claims and beyond from the metrics avialable:
>
> Environment-based (Allelopahtic Harvest):
> - ripe berries by type
> - unripe berries by type
> - berries by type
> - coloring by player
> - eating types by player
> - berries per type by color of colorer
> - berries per type by taste of colorer
> - player timeout count
> - color by color zap counts
> - color by taste zap counts
> - taste by taste zap counts
> - taste by color zap counts
> - who zapped who
>
> Algorithm based (RLlib PPO - used by you for training):
> - policy loss
> - value function loss
> - total loss
> - entropy
> - Kullback-Leibler (KL) divergence
> - learning rate
> - clip fraction
> - approximate Kullback-Leibler (approx_kl)
> - explained variance
> - timesteps per second
> - rewards
> - ...
>
> And finally, custom metrics that you might have been tracking to support your claims in regards to your mechanism. As you have provided:
> - Mean Episode Reward
> - Adjusted Mean Episode Reward (This is still named incorrectly, against your claims of having this changed, in your latest revision in Figure 2, adding to continued confusion).
> - Correct Sanctions Fraction
> - Incorrect Sanctions Fraction
> - ...
>
> I don't think I'm asking too much for authors to provide full result metrics on their experiments.
>
> I have tried to go deeper into your code, but there is no README.md provided to explain what exactly has been done to reproduce your results. Furthermore, I have realized that you were right. I do not fully understand the background of this work, but it is incredibly hard to understand all the implementation details of your mechanisms. I will adjust my rating to reflect this.

---

> > ### Author Response · Authors · 2024-12-02
> > **Response to Reviewer Dy2Z**
> >
> > We have provided comprehensive responses to the reviewer's original questions and explained how our reported metrics support the claims in the paper, yet the reviewer continues to avoid engaging with our detailed explanations. While we acknowledge one could report every possible random metric, the paper focuses on those that are relevant to and support its claims. The additional metrics requested for Allelopathy (implemented as debug variables in Melting Pot, and not reported in their original paper) and generic RLLib PPO metrics are not pertinent to our claims. Given that these "fine-grained" metrics were not mentioned at all in the initial review, this represents a significantly big and unnecessary ask at this stage.

---

### Official Review · Reviewer_HNPm · 2024-10-29

**Soundness:** 1
**Presentation:** 1
**Contribution:** 2
**Rating:** 3
**Confidence:** 4

**Summary:**

This paper proposes tackling the alignment challenge in multi-agent systems using normative classification institutions. They consider two socially challenging multi-agent reinforcement learning environments augmented with normative infrastructure, which they call “altars”. These altars are features of the environment associated with functions that determine whether agent sanctioning actions are “right” or “wrong” in Sanction-Augmented Markov Games. They investigate the effect of such infrastructure on the learning dynamics and alignment of agents in their environments.

**Strengths:**

1. Cooperative alignment in multi-agent systems, the problem this paper aims to tackle, is highly relevant and important.
2. The idea is good. Classification institutions for normative alignment make intuitive sense and are well-grounded in social science and anthropology.
3. The introduction is very well written.

**Weaknesses:**

1. To me, Figure 1 is not very informative. This figure should provide a clear and unambiguous overview of your approach for the busy reader. The caption is just “Overview of the Altared Framework”. What are the three subfigures? What actually is your framework and how is it depicted here? What is “history”? I feel like this could be made clearer.

2. Section 3 does not flow well and feels confused. There is very low information density per word here. I feel there are enough issues to warrant a full section-by-section breakdown:

    3.1. introduces the problem and how you fix it. This is great.

    3.2 defines I-SMG. Is this the framework you propose? Figure 1, the central figure of the paper, mentions an “altared framework” but this is not defined anywhere. I would expect a description of your core proposal here.

    3.2.1 “salient features of an institution”. I am not sure what this adds. The “Prescriptive” sub-sub-sub-section seems to be additional explanation of I-SMG. This is helpful but out-of-place and presented confusingly. The “Dynamic” points range from vague (what is avg_behaviour? what is an “annual policy update function” and why would we apply the same transformation every year?) to outright confusing:

    - 1. What do tax policies and insurance regulations have to do with social norms? You explicitly mention that the institutional prescription does not modify rewards, but it seems to me that taxation would.

    - 2. How do we go from \tau(I_t, s_t) to averaging over the past k states? The only explanation we get is “where h adjusts the institution based on recent agent behaviors”. This is very vague and I suspect violates the Markov property of the I-SMG.

    3.3 this section finally introduces the word “altar” again, but honestly I have no idea what it adds. Everything in here is just a repeat of things that have already been said, and there is no reference to I-SMG to tie it all together.

    3.3.1 and 3.3.2 - environment descriptions. These are long sub-sub-sections of the proposed method section and do not flow well from it. Why are they not in “4.1 Environments”? Importantly, I do not like that these sections do not give a self-contained description of the environments but instead point the reader to the appendix. When reading a paper, the first thing I personally want to do is check results; to do this, I should be able to quickly get a nice concise explanation of the test environments and then move on - not scroll down to the appendix for such crucial information. Finally, we go back to an explanation of why/how your method works at the bottom of 3.2.2 - this again feels out of place and does not flow well (in what way is this specific to the Allelopathic harvest environment?).

3. Section 4 is similarly poorly written. A non-exhaustive list of comments:
    1. "4.1 Environments" is just one short uninformative paragraph. Is this half written? It says “we describe the variations here” but does not do so. This seems a good place for the environment descriptions above…
    2. "4.2 Baselines and Agent Architecture” and then "4.2.1 Agent Architecture” and no other sub-sub-sections. Was this proofread?
    3. Why are environment descriptions, needed to understand your results, in the appendix but agent architecture in the main paper?
    4. Are the plots just WandB screenshots? I would expect much neater plots with axis labels and much longer, descriptive figure captions from a paper submitted to ICLR. Why are some lines truncated earlier than others? Font size is a bit small too.
4. Weak results. This is my primary concern and biggest reason I’m going to recommend rejection. These results are far from convincing. It looks to me like there is no difference at all in mean episode reward for allelopathic harvest, and three seeds is not enough for a rigorous evaluation on Commons Harvest. I have worked with the meltingpot environments myself in the past and have seen firsthand just how stochastic these are. Even more concerningly, the error bands seem to abruptly shrink twice on your Commons Harvest mean episode reward plot, with no variance at all at the end; this looks a lot like the three seeds ran were not all run to termination. Unless you can explain these two sharp decreases in variance, it seems to me one of the core plots supporting your approach is presenting misleading information.
5. No ethics section? Who gets to decide the norms that the institution classifies as “right”?
6. (very minor) latex comments:
    1. some of the notation in definitions is unnecessarily verbose - I personally prefer \bigtimes for long cartesian products.
    2. Sanction function being called Sigma is ambiguous given this is summation notation, e.g. ambiguity in SMG reward function definition.

**Questions:**

Please address the questions raised above - especially those concerning the experimental results.

---

> ### Author Response · Authors · 2024-11-29
> **Response to Reviewer HNPm (1/2)**
>
> We are thrilled to hear that the reveiwer finds our overall approach very good, intuitive, and well-grounded. Your thoughtful and detailed review, particularly your extended efforts in identifying presentation flaws, is greatly appreciated. We sincerely apologize for the shortcomings in our initial writing. Your feedback has been invaluable and has motivated us to significantly improve the presentation of our work, elevating it by an order of magnitude. Below we address your specific concerns:
>
> > Issues with clarity, flow and presentation of Section 2, Section 3 and Section 4
>
> As mentioned in our general comment, we have revamped these three sections in response to your valuable feedback. We have streamlined them with a strong emphasis on clarity, accessibility, and accuracy, ensuring they are significantly improved and better aligned with the goals of the paper.
>
> Specifically, we have worked to ensure consistent definitions and references across all sections, including the experiments and corresponding plots.
> - In **Section 2**, we introduce the theoretical framework that underpins our methods, provide the formalism of the Sanction Augmented Markov Game (SMG), and discuss how this framework serves as the foundation for the Hidden Rule SMG baseline used in our experiments.
> - In **Section 3**, we address the key challenge of dynamic norms in our setup, offering a formal discussion of this issue. We then present one of our main contributions: the extension of Markov Games to their altered versions, referred to as Altared Games. We do not use the term I-SMG anymore in our paper and also rectified any inconsistencies and redundancies around its description.
> Throughout these sections, we have ensured consistent referencing of all methods.
> - We have provided a detailed explanation of the core mechanics of the environments in Section 4.1. Additionally, we discuss the three experimental conditions tested in our work in Appendices A.1 and A.2. As noted earlier, we would ideally like to bring Appendix A to the main paper but for now, we have updated the paper for full clarity on our proposed approach, related background and empirical investigation. We will reorganize further to get it in main paper in the final version.
>
> > Improvements to Figures:
>
> We acknowledge that Figure 1 is not very informative in the earlier draft. We have updated its caption to better explain our framework and are in the process of improving the figure itself, which will be included in the final version of the paper. We have also revised the figure captions throughout the paper to ensure they are more informative and descriptive. Finally, we have corrected the plot colors and fixed figure references throughout the paper to ensure visual and contextual consistency.
>
> > No ethics section? Who gets to decide the norms that the institution classifies as “right”?
>
> Thank you for raising this concern. To answer your question, norms in multi-agent systems can originate in several ways. They may be exogenously provided, pre-defined by system designers or external entities, and encoded into the environment to represent ideal behaviors. Alternatively, norms can emerge from interactions among agents, evolving organically through repeated interactions and collective decision-making processes. Both approaches have been extensively explored in fields such as sociology, political theory, and artificial intelligence. However, this study does not aim to explore the origin or establishment of norms, nor does it investigate how rules should be designed or who determines their content.
>
> We assume that a normative system is already in place, with predefined but dynamic norms (as discussed in Section 3) and associated enforcement mechanisms. Our focus is on investigating the role of explicit institutional mechanisms, such as the altar, in improving agents' ability to learn and adapt to these evolving norms, particularly in dynamic environments. To the best of our knowledge, the role of institutions in supporting norm compliance for AI agents has not been studied before. Hence, we focus on demonstrating that an ideal normative system produces higher payoffs when agents have access to institutions that enhance their capacity to learn and track the content of norms. This allows agents to adapt more rapidly and reliably to a changing normative structure, irrespective of how those norms originated. While broader questions regarding the emergence and establishment of norms are beyond the scope of this work, readers can refer to relevant literature, such as [1,2] for a detailed discussion.
>
> We will include this discussion in the final version of the paper.
>
> [1] A survey of research on the emergence of norms in multi-agent systems, Bastin Ton Roy Savarimuthu and Stephen Cranefiled, 2011.
>
> [2] Norm emergence in multiagent systems: a viewpoint paper, Andreasa Morris-Martin, Marina De Vos and Julain Padget, AAMAS 2019.

---

> > ### Author Response · Authors · 2024-11-29
> > **Response to Reviewer HNPm (2/2)**
> >
> > > Weak results. This is my primary concern and biggest reason I’m going to recommend rejection. These results are far from convincing. It looks to me like there is no difference at all in mean episode reward for allelopathic harvest, and three seeds is not enough for a rigorous evaluation on Commons Harvest. I have worked with the meltingpot environments myself in the past and have seen firsthand just how stochastic these are. Even more concerningly, the error bands seem to abruptly shrink twice on your Commons Harvest mean episode reward plot, with no variance at all at the end; this looks a lot like the three seeds ran were not all run to termination. Unless you can explain these two sharp decreases in variance, it seems to me one of the core plots supporting your approach is presenting misleading information.
> >
> > We sincerely thank the reviewer for their detailed feedback and for highlighting the issues with the variance in the Commons Harvest plot. Upon closer inspection, we discovered an oversight: one of the seeds in the report came from a different batch of runs and had terminated well before the others, resulting in the sharp drop in variance observed. We deeply apologize for this mistake and have since corrected it. The updated plot now reflects only runs that were completed, and we have ensured consistency across all presented results.
> >
> > Further, we fully agree with the reviewer that running experiments on only three seeds is insufficient for a rigorous evaluation. This limitation was primarily due to computational constraints during the initial submission. However, based on your valuable feedback, we have now extended our experiments to five seeds, and we commit to expanding this further to ten seeds in the final version of the paper. This will provide a more robust evaluation of our approach.
> >
> > Please refer to **Figure 2** and **Figure 3** in the revised manuscript. These figures now present the results in a concise and precise manner. As clarified in the updated manuscript, instead of raw mean episode reward, we should be looking at the Adjusted Mean Reward for both the Hidden Rule SMG and Altared SMG setups. This adjustment excludes the artificial positive rewards gained from correct sanctioning, which are designed to train agents, and instead includes:
> > Adjusted Mean Reward = Base environmental rewards (e.g., consuming a berry or harvesting an apple). - the cost of sanctioning - the cost of being sanctioned.
> >
> > This adjustment ensures that the reported rewards reflect meaningful cooperative performance by focusing on sustained outcomes. It demonstrates that the sanctioning mechanism is beneficial only when the environmental rewards outweigh the costs incurred through sanctioning, providing a fair and transparent comparison.
> >
> > For the Commons Harvest environment, our updated results continue to demonstrate strong performance compared to the baseline without institutions. Importantly, the updated results show consistent trends across all seeds, addressing the issue of variance previously noted.
> >
> > For the Allelopathic Harvest environment, we observe improved performance for our approach, and, more critically, a reduction in variance over training time. This reduction in variance suggests that agents adapt more effectively to dynamic norms when institutions are incorporated into the framework. However, we acknowledge that Allelopathic Harvest is a particularly slow environment, and the runs have not yet reached full termination. We are actively continuing these experiments and will share the final plot for this environment before the end of the discussion period. That said, even with the current progress, the variance observed in the hidden rule baseline (~~200) baseline approach remains significantly higher than in our method (~40), underscoring the robustness of our framework.
> >
> > We hope that the updates we have provided, particularly regarding the presentation, experimental results, and empirical reporting, comprehensively address the concerns raised. We kindly invite the reviewer to revisit their score in light of these clarifications and improvements. If there are any remaining questions or areas where further discussion would be helpful, we would be delighted to engage further to ensure all concerns are fully addressed.

---

### Official Review · Reviewer_oEzP · 2024-10-31

**Soundness:** 2
**Presentation:** 1
**Contribution:** 2
**Rating:** 3
**Confidence:** 3

**Summary:**

This work tackles the problem of collaboration in mixed-motive multi-agent reinforcement learning. The authors suggest introducing an addition to the agents' observations, the altar, that indicates certain institutional rules that the agents should follow to maximize social welfare. This is in contrast to previous work in which these institutional rules are hidden. Empirical results over two grid-based domains show that agents trained with altared versions of the environment achieve higher rewards and are resilient to changes in these institutional rules.

**Strengths:**

- The idea is novel and deals with an interesting problem in multi-agent AI research.
- Results show that indeed altared environments promote collaboration and resilience in agents.

**Weaknesses:**

- Poorly written and difficult to read:
	- There are many grammatical errors that lead to ambiguity and ruin the flow of the text. It is  recommended to use automated grammar checking tools as this is present throughout the entire text.
	- Paragraphs are long and packed with multiple ideas making them hard to read. Each paragraph should capture one main idea.
	- There are many run-on sentences which are difficult to follow. Shorter, more concise sentences will be easier to follow.
	- There are broken links to appendices, unreferenced figures, and overall messy presentation of results.
- The authors claim to have published their code for reproducibility, but following the link provided in the abstract leads to an empty repository (https://anonymous.4open.science/r/normarl-public-804C/README.md).
- The authors make claims and mentions models that are not explained nor cited. A glaring example from the introduction is in line 45: "literature in multi-agent systems (MAS) has grappled with...". This is present several other places as well. Another example is the lack of citation for definition of models in the background, e.g., Markov game (line 107). The authors should either cite, prove, or give an intuitive explanation to every claim in the paper.
- The authors define many decision-making frameworks in section 2 and sections 3.1, 3.2, but these are not revisited in the main parts of the paper which makes them confusing and redundant.
- The contribution of this work is can be boiled down to adding manually programmed indications of social rules explicitly into the agents' observations. This is done with no theoretical justification and relatively weak intuition. You should either mathematically prove that using the altar modification encourages correct sanctioning, or at least give a stronger intuition as to why this effect might occur.
- Figure 1, is not clear at all and does not help demonstrate the differences between the  versions of Markov games. Also, it appears at the top of page 2 with an uninformative caption, but is first referenced on page 4. Figures should be near where they are first referenced because otherwise the reader is forced to constantly scroll and break concentration.
- Figure 2 is not referenced anywhere in the text.
- Environment description in main paper is insufficient. e.g., It is mentioned in line 315 out of the blue that the agents can eat the berries.
- showcases only harvest domains. There are other social dilemma domains like cleanup, economist, prisoners dilemma, etc. Every social dilemma environment presents its own challenges that can be potentially be solved by smart sanctioning. Testing in these environments could make a stronger case for using altars or might provide insights into the limitations of altars.
- There is inconsistent plot coloring and labeling of plots, making it difficult to follow the results.

**Questions:**

- why is POMG mentioned? are the environments partially observable? if so, this is not clear from the text.
- Why is I-SMG defined? it seems redundant. Isn't it enough to define an altar and explain that it induces institutional rules? what is the difference between an altar and an I-SMG institution?
- In figure 6 the authors show 6 plots. why does figure 7, which is essentially the same kind of results but for a different environment, present only 3?

---

> ### Author Response · Authors · 2024-11-29
> **Response to Reviewer oEzP (1/3)**
>
> We thank the reviewer for recognizing the novelty and strength of our results. Thank you  for pointing out the presentation issues which have certainly helped to improve the quality of the paper and we have fixed all of them in the revised version of the paper.
>
> > The contribution of this work is can be boiled down to adding manually programmed indications of social rules explicitly into the agents' observations. This is done with no theoretical justification and relatively weak intuition. You should either mathematically prove that using the altar modification encourages correct sanctioning, or at least give a stronger intuition as to why this effect might occur.
>
> We respectfully disagree with the reviewer’s comments on a fundamental level.
>
> The key objective of this work is to draw inspiration from models of human societies to investigate whether a normative institution can improve the capacity of architecturally simple AI agents to adapt to dynamic norms while interacting in mixed motive environments. We are ultimately interested in demonstrating that the human challenge of aligning AI agents with our preferred outcomes could be better solved by expanding our efforts beyond algorithm design to environment design. To conduct this investigation, we systematically design our approach, each of which represents a preliminary but foundational step towards these broader research questions on building scalable, adaptable, and socially aligned AI systems.
>
> First, we ground our formal approach in the theoretical framework of the rational agent model of normative social order. We adopt this framework in the context of AI research, extending Markov games to include an explicit environmental feature called the altar—representing an authoritative focal point that articulates a group’s shared norms or laws. We then tackle the challenging setup of dynamic norms, providing both a recipe and an example to convert a given Markov game into its Altared version. To empirically investigate our framework, we employ a methodology of controlled hypothesis testing, inspired by social science, to isolate the role of the altar. This approach ensures that the effects of the altar can be assessed independently by holding all other factors constant across experimental setups. The manually programmed indications of social rules, as mentioned by the reviewer, are a deliberate design choice to enable this controlled testing and not a weakness of the work (see our detailed response to reviewer BYmR). As reviewer HNPm correctly pointed out, our formal approach and experimental design are deeply rooted in theoretical frameworks from social science and anthropology, supported by strong real-world intuitions and examples from human societies. Finally, we demonstrate that explicit classification institutions, such as the altar, improve agents’ ability to cooperate, enforce norms accurately, and adapt effectively to uncertainty.
>
> > The authors claim to have published their code for reproducibility, but following the link provided in the abstract leads to an empty repository
>
> We apologize for the oversight regarding code accessibility. The code has now been made available at the same link, enabling further investigation of our work.
>
> > why is POMG mentioned? are the environments partially observable? if so, this is not clear from the text.
>
> Yes, both the environments are partially observable and the agents only have access to their cloak observation. We have described the details on the observations in Appendix E.
>
> > In figure 6 the authors show 6 plots. why does figure 7, which is essentially the same kind of results but for a different environment, present only 3?
>
> Please refer to **Figure 2** and **Figure 3** in the revised manuscript. These figures now present the results in a concise and precise manner. Previously, Figure 6 included two rows, but the second row did not display meaningful data. Specifically, the bottom row in the earlier version reported raw Mean Episode Rewards. However, as clarified in the updated manuscript, we should be looking at the Adjusted Mean Reward for both the Hidden Rule SMG and Altared SMG setups. This adjustment excludes the artificial positive rewards gained from correct sanctioning, which are designed to train agents, and instead includes:
> - Base environmental rewards (e.g., consuming a berry or harvesting an apple).
> - The cost of sanctioning.
> - The cost of being sanctioned.
>
> This adjustment ensures that the reported rewards reflect meaningful cooperative performance by focusing on sustained outcomes. It demonstrates that the sanctioning mechanism is beneficial only when the environmental rewards outweigh the costs incurred through sanctioning, providing a fair and transparent comparison.

---

> > ### Author Response · Authors · 2024-11-29
> > **Response to Reviewer oEzP (2/3)**
> >
> > > Poorly written text, claims and models missing citation and explanations, lack of clarity in Section 2 and Section 3, Problems with figures, plots and environment descriptions
> >
> > We thank the reviewer for highlighting these presentation issues. As part of the revision process, we have made the following updates in response to your specific points:
> >
> > - **Clarifications and Proofreading:**
> > We have revised the writing to clarify any potentially misleading details, thoroughly checked for errors, and proofread the text to address any missing citations or references. We have also ensured that all claims originating from previous literature are appropriately cited. However, we welcome further feedback from the reviewer if any aspect has been overlooked.
> >
> > - **Streamlining Sections:**
> > We have streamlined Sections 2, 3, and 4 to ensure consistent definitions and references across all sections, including the experiments and corresponding plots. Specifically:
> >   - In **Section 2**, we introduce the theoretical framework that underpins our methods, provide the formalism of the Sanction Augmented Markov Game (SMG), and discuss how this framework  serves as the foundation for the Hidden Rule SMG baseline used in our experiments.
> >   - In **Section 3**, we address the key challenge of dynamic norms in our setup, offering a formal discussion of this issue. We then present one of our main contributions: the extension of Markov Games to their altered versions, referred to as Altared Games. We do not use the term I-SMG anymore in our paper and also rectified any inconsistencies and redundancies around its description.
> >   - Throughout these sections, we have ensured consistent referencing of all methods.
> >
> > - **Improvements to Figures:**
> > 	We acknowledge that Figure 1 is not very informative in the earlier draft. We have updated its caption to better explain our framework and are in the process of improving the figure itself, which will be included in the final version of the paper.
> >
> > - **Environment Mechanics and Conditions:**
> > We have provided a detailed explanation of the core mechanics of the environments in Section 4.1. Additionally, we discuss the three experimental conditions tested in our work in Appendices A.1 and A.2.
> >
> > - **Visual Consistency:**
> > We have corrected the plot colors and fixed figure references throughout the paper to ensure visual and contextual consistency.
> >
> > We hope these changes address the reviewer’s concerns and significantly improve the clarity and presentation of our work. We remain open to additional suggestions to further refine the paper.

---

> > > ### Author Response · Authors · 2024-11-29
> > > **Response to Reviewer oEzP (3/3)**
> > >
> > > > showcases only harvest domains. There are other social dilemma domains like cleanup, economist, prisoners dilemma, etc. Every social dilemma environment presents its own challenges that can be potentially be solved by smart sanctioning. Testing in these environments could make a stronger case for using altars or might provide insights into the limitations of altars.
> > >
> > > Thank you for your suggestion. Given an environment with well-defined normative challenges, the key objective of our work is to investigate the role of explicit institutional mechanisms, such as the altar, in improving agents' ability to learn and adapt to evolving norms in dynamic environments. We chose these two environments as they highlight diverse challenges of aligning individual incentives with group goals and provide a rich domain for exploring the role of norms and enforcement mechanisms. Although both are called "harvest" games, they present fundamentally different social dilemmas for groups to solve.
> > >
> > > Commons Harvest captures the dynamics of the tragedy of the commons, where agents must sustainably manage shared resources to prevent catastrophic depletion.  Agents in this environment harvest apples from a shared orchard. Regeneration of apples is a function of the amount of apples in a local neighborhood. Individual agents earn rewards from harvesting but harvesting imposes an externality on other agents by reducing regrowth rates. These individual incentives can learn to overharvesting and potential collapse of the orchard.  Agents are able to generate group benefits (higher collective payoffs) if they can implement norms that constrain harvesting to sustainable levels.
> > >
> > > In Allelopathic Harvest, agents make decisions about whether and which of three possible crops (colored berries) to plant and which to eat; agents have divergent individual payoffs associated with each berry color (some prefer red, some prefer green, and all prefer red or green to blue). The growth rate for berries, however, is highest if all agents plant the same color (ie. plant a monoculture). This game thus involves equilibrium selection (which color to plant) and free-riding problems (agents would prefer to eat and not plant). Agents in this environment generate group benefits if they can implement norms that coordinate the agents on planting the same color berries and limit free-riding.
> > >
> > > Finally, regarding some of the environments suggested by the reviewer, we would like to highlight that the environments selected
> > > in this work either subsume key properties of those suggested and introduce more interesting and realistic challenges. For instance, the Allelopathic Harvest environment incorporates both the delayed reward effect (as planting yields no immediate reward) and the free-rider problem—core aspects of the Cleanup environment. Similarly, the Commons Harvest environment captures a compelling intertemporal social dilemma, where decisions that are optimal for individuals or groups in the short term may result in suboptimal outcomes in the long run. While the paper already provides a framework to create altered versions of such environments, we greatly appreciate the reviewer’s suggestion. It has inspired us to work on creating and releasing a comprehensive suite of environments with varied normative setups. This dataset will aim to facilitate further research within the community. We will include this as part of the final version of the paper if accepted.
> > >
> > > We believe that the updates and responses we have provided address the key concerns raised. We kindly invite the reviewer to revisit their score in light of these clarifications and improvements. If there are any remaining questions or areas where further discussion would be helpful, we would be delighted to engage further and ensure all concerns are fully addressed.

---

### Official Review · Reviewer_BYmR · 2024-11-04

**Soundness:** 3
**Presentation:** 3
**Contribution:** 2
**Rating:** 5
**Confidence:** 4

**Summary:**

The paper describes the importance of cooperation to reach socially beneficial equilibriums in mixed motive environments, which often present a variety of cooperation challenges. Specifically, this paper mentions the tragedy of commons, alignment, and the free-rider problem. They use the Allelopathic Harvest and Commons Harvest environments, making modifications as needed.
For the purposes of this study the agents used to explore these concepts use multi-agent reinforcement learning, specifically using PPO in self-play. However, this is not the focus of the study, and the specific learning method is not the primary focus and should be viewed as such. They claim to be agnostic to the learning method and have further claims that similar performance with and without GRU units.

This is due to the authors want the study to not focus on additions to the model of the agent, but rather on changing the environment to affect the norms. They wish to understand institutions and describe multiple formulations of institutions in the work.

The authors show they are wanting to study the effect of introducing “altars” or institutions into the environment (which can act similarly to the formulations they have described, ie be dynamic, fixed, etc…) to show that this can help solve the alignment problem. They do this by explicitly adding altars to the map, which are zones which prescribe the colour of the resource that should be used.

Many variants of the game are described, however the variants of the games which are most applicable are those in which the agents can use some method to tag/sanction their fellow agents to encourage desired social norms. These tagging abilities have different rewards based on a hidden rule or the altar colour in the environment. In some versions distraction institutions are added and the agents are still capable of learning.

The presence of the altar is shown to have a direct effect of increasing the total reward (adjusted for hidden rewards) of the agents even as compared to the hidden reward, which are both capable of suggesting the same norm for the behaviour.

**Strengths:**

The work is well presented, and easy to follow.

The experiments do show that the introduction of an altar does reduce the cognitive burden on agents while addressing cooperation challenges and promote normative social order. And that performance is increased compared to simply having a hidden rule/norm built into the reward of the sanctioning actions.

The experiments are solid and show the improvement achieved by the introduction of altars.

**Weaknesses:**

These institutions/altars have to know the environment rules to dynamically change. I am not convinced we always have that knowledge available, especially when it is environment dependent (like when it has to do with the relative scarcity of the apples/berries in different zones).

It would also have to have some sense of the possible norms before it can be introduced and then we're simply choosing one for it to represent.

That brings in the introduction of an altar with dynamic rules, which has to match the reward structure of the tagging. Is this not something we have to know before hand? Or is that seen as dictated by the environment. If it is, how do we enforce the altar to follow those rules? We won't always know them like we do in these situations. Is there a possibility of them having to learn how to change to fit the environments rules? (How will its observations work then?)

We won't always be able to alter an environment to include an altar, how do we address such situations?

Do these altars not have to have the reward structures with regards to tagging hard programmed into them to follow the rules to show agents the correct representation colour?

**Questions:**

Please see the strengths/weaknesses.

However, see below, I am open to debate on these and willing to be persuaded to adjust my score, I just have to be convinced after the following questions:

1. Does the introduction of the altar not add a new observation data point into the input? Or is this just directly learned over time and from passing it in earlier states? I am assuming the latter?

2. Do these altars not have to have the reward structures with regards to tagging hard programmed into them to follow the rules to show agents the correct representation colour? (is there a possibility of them having to learn how to change to fit the environments rules?)

3. With the introduction of an altar with dynamic rules, which has to match the reward structure of the tagging. Is this not something we have to know before hand? Or is that seen as dictated by the environment. If it is, how do we enforce the altar to follow those rules? We won't always know them like we do in these situations. Is there a possibility of the altar having to learn how to change to fit the environments rules? (How will its observations work then?)

4. We won't always be able to alter an environment to include an altar, how do we address such situations? Or is this simply not applicable then?

5. How do we get a sense of the available norms before being able to add the altars?

---

> ### Author Response · Authors · 2024-11-29
> **Response to Reviewer BYmR (1/2)**
>
> We thank the reviewer for recognizing the strength of our experiments and that they show solid evidence of the hypothesis we sought to test, namely that the introduction of a normative institution–the ‘altar’ which represents the state of a dynamic ‘rule’ for the agents–improves the average reward achieved by the agents. We note that the reviewer’s description of the strengths of the paper fully captures our goal and contribution, although the summary of the paper is fairly confusing and may mislead someone who has not read the paper.
>
> We are encouraged that the reviewer says they are open to being persuaded to adjust their score because the weaknesses that concern the reviewer stem from a misunderstanding of our aims and the nature of our contribution. The reviewer appears to believe that our primary goal is to demonstrate improved performance in the Commons Harvest and Allelopathic Harvest environments through tagging (punishing) according to rules. Given this mistaken belief, the reviewer is appropriately concerned about where the rules come from. However, this is not the objective of our work. As prior literature (e.g., Vinitsky et al., Koster et al.) has already shown, MARL agents can learn to punish according to predefined rules, and when these rules incentivize cooperation—such as coordinating to plant same color in Allelopathy or reducing over-harvesting in Commons Harvest—then this punishment improves performance.
>
> Our focus, instead, is on demonstrating that given the right rule and a reward structure incentivizing agents to punish violations of that rule, agents are able to improve their ability to learn and implement the rule–achieve better performance in the face of a social dilemma– when the environment includes an institutional feature (the altar) representing the rule, compared to when it does not. This is why we compare two specific conditions: (1) the baseline hidden rule setup (Hidden Rule SMG), and (2) the altar-augmented setup (Altared SMG). The reward structure for punishment, which is controlled by the experimenter, is identical across both conditions, ensuring that the comparison isolates the effect of the altar.
>
> Hence we are testing only the impact of the presence of a feature that represents the rule. We are not testing the impact of the content of the rule but rather controlling it to conduct a valid experimental test of our hypothesis: that the presence of the altar feature itself causes improved performance. Our approach aligns with social science methodologies, which emphasize controlling all other sources of variability to isolate the causal impact of the variable of interest. This design allows us to rigorously test the hypothesis that explicit institutional representation enhances agents' ability to learn and enforce norms, ultimately improving outcomes in social dilemmas. (This is an experiment to test the impact of this environmental feature, not a more conventional experiment to show improved performance from a new algorithm compared to benchmark performance.) Our approach aligns with social science methodologies, which emphasize controlling all other sources of variability to isolate the causal impact of the variable of interest.
>
> We think this experimental design has been misunderstood in our paper and we have revised our exposition to help ensure readers do not make this misinterpretation of our work. We now address your specific questions in order:
>
> > Does the introduction of the altar not add a new observation data point into the input? Or is this just directly learned over time and from passing it in earlier states? I am assuming the latter?
>
> Yes, the altar adds a new observation and this is precisely what we show: with this informational enrichment of the environment, agents are able to better learn the structure of punishment rewards. In the baseline condition, agents have to learn what punishment actions earn a reward solely from observing the environment and tracking how their actions produce rewards. In the altar condition, they have all this information plus an additional observation: a color on the altar. What we are showing is that adding this observation makes the learning problem easier and, moreover, that the agents learn to visit the altar in order to obtain this observation. We make the content of the altar dynamic so as to demonstrate this effect.

---

> > ### Author Response · Authors · 2024-11-29
> > **Response to Reviewer BYmR (2/2)**
> >
> > > Do these altars not have to have the reward structures with regards to tagging hard programmed into them to follow the rules to show agents the correct representation colour? (is there a possibility of them having to learn how to change to fit the environments rules?)
> >
> > Yes, the altars have the reward structures hard programmed, in exactly the same way that the reward structures are hard programmed in the baseline (hidden rule) condition. As explained above, this is the requirement for a valid experimental test that the introduction of an altar, as an information feature of the environment, causes improved behavior. We are controlling the content of the reward structure in both conditions to ensure it does not vary. We are not testing the impact of alternative reward structures for punishment; we are testing the impact of having the reward structure represented as a feature. We are comparing environments, not rules.
> >
> > > With the introduction of an altar with dynamic rules, which has to match the reward structure of the tagging. Is this not something we have to know beforehand? Or is that seen as dictated by the environment. If it is, how do we enforce the altar to follow those rules? We won't always know them like we do in these situations. Is there a possibility of the altar having to learn how to change to fit the environment's rules? (How will its observations work then?)
> >
> > We hope that the answer to this question is now clear: yes, the altar representation has to match the reward structure of tagging because we are not evaluating alternative rules, we are only testing the impact of having an institutional representation of the rule. This is necessary for a valid test of our causal hypothesis. This paper is a preliminary but foundational step for building AI systems in a broader research agenda that explores questions like how an altar (e.g. the announcements of a leader or court) generates rewards that incentivize and coordinate punishment behavior (a theoretical question analyzed in Hadfield & Weingast 2012), which we cite as the grounding for our work), and the very broad question of how the content of rules is established and evolves in dynamic environments. It might be helpful to think of the altar as representing the human-controlled rules (arrived at through human processes like democratic legislatures or regulators) and punishment rewards for agents being determined by the content of the altar in order to incentivize agents to punish in accordance with the human-controlled rule.
> >
> > > We won't always be able to alter an environment to include an altar, how do we address such situations? Or is this simply not applicable then?
> >
> > We are ultimately interested in demonstrating that the human challenge of aligning AI agents with our preferred outcomes (e.g. not depleting a shared resource, coordinating on a particular equilibrium like planting a specific type of crop) could be better solved by expanding our efforts beyond algorithm design to environment design. We’re not sure what the obstacles to adding an ‘altar’ into an environment might be, but yes, if there are environments where this is not possible then an implication of our work is that we should expect poorer performance from agents, even when we supply them with (implicit, hidden) desired punishment incentives.
> >
> > > How do we get a sense of the available norms before being able to add the altars?
> >
> > We hope the answer to this question is also now clear: we are not studying alternative rules/norms; we are studying the features of the environment in which norm content is learned by agents. We have designed our study to test our hypothesis: that assuming a group has discovered the ideal cooperation-promoting norm, how can the benefits of this norm be best achieved? We show that an ideal norm produces higher payoffs when the agents have an institution that improves their capacity to learn and track the content of the norm and so more rapidly and reliably adapt their behavior to a changing ideal norm.
> > While broader questions regarding the emergence and establishment of norms are beyond the scope of this work, we refer the reviewer to relevant literature [1,2]  for a detailed discussion. Further, while we discuss this briefly in our extended related work section in Appendix B, we will also add a discussion section specifically highlighting this question.
> >
> > [1] A survey of research on the emergence of norms in multi-agent systems, Bastin Ton Roy Savarimuthu and Stephen Cranefiled, 2011.
> >
> > [2] Norm emergence in multiagent systems: a viewpoint paper, Andreasa Morris-Martin, Marina De Vos and Julain Padget, AAMAS 2019.
> >
> > We believe that our responses address the key concerns raised. We kindly invite the reviewer to revisit their score in light of these clarifications and improvements. If there are any remaining questions or areas where further discussion would be helpful, we would be delighted to engage further and ensure all concerns are fully addressed.

---

### Author Response · Authors · 2024-11-29
**Global Response on Strengthening Clarity, Presentation and Evaluations**

We sincerely thank all the reviewers for their thoughtful and detailed feedback.

We are glad to see that the reviewers appreciated several aspects of our work, including its novelty and relevance for multiagent AI research, strong intuitions and grounding in social science and anthropology and strength of our empirical investigation.

The reviewers also identified opportunities for improvement, especially highlighting several misunderstandings, mostly created due to the lack of clarity in the presentation of both our formal model and description of our empirical setup. We are very grateful for all the constructive comments that have certainly pushed us to improve our articulation of the work and helped to enhance the quality of our paper by orders of magnitude.

Based on the feedback, we have made substantial updates in the revised paper, which we briefly summarize here and discuss in more detail in individual responses to each reviewer.

**Clarified Intuitions, Formal Model and Setup**

- While most reviewers appreciated our introduction, we recognize that some of the misunderstandings reflected in the reviews may stem from how we initially articulated certain points. To address this, we have revised the introduction for greater clarity and accuracy.
- We have updated Section 2 to include a more detailed discussion of the theoretical framework and provide a cleaner formal description of our baseline—the Hidden Rule Sanction-Augmented Markov Game (Hidden Rule SMG).
- We have updated Section 3 to first discuss the formal setup of dynamic norms, followed by a revised and clearer presentation of our proposed approach: the Altared Sanction-Augmented Markov Game (Altared SMG or Altared Games, previously referred to as I-SMG)
- We have refined the entire manuscript to emphasize the key rationale behind our experimental setup. Our primary objective is to investigate the impact of institutions (the altar) on trained agents' normative behavior compared to environments without institutions. We conduct a controlled experiment to accomplish this objective, maintaining the same norms in both experimental conditions. This allows us to isolate the causal effect of introducing an altar. That is to say: **this work focuses on comparing environments, not AI algorithms**.
- As we now make clearer in the paper, similarly, **this work does not focus on the selection, establishment or emergence of the content of norms**, which is a distinct problem. We are not testing the capacity of the agents to identify group-beneficial norms. We emphasize that the capacity to maintain a normative social order--coordinating third-party enforcement of norms--is distinct from the capacity to select group-beneficial norms, and is likely governed by very different types of processes.  This is a key result from the cultural evolution literature for example, where [1] first showed an evolutionary game context that third-party punishment "allows the evolution of cooperation (or anything else) in sizable groups." The selection of group-beneficial norms is likely governed by group-selection processes and thus must be studied by evaluating relative group performance. This is out of scope for this work, where we hard-code group-beneficial norms and then evaluate whether the presence of an altar improves the capacity of groups to achieve those norms and hence those benefits.

**Improvements to Figures and Writing**

- We have revised the writing to ensure precision and conciseness in our exposition, eliminating any potentially misleading statements that may have contributed to misunderstandings. Additionally, we have thoroughly proofread the manuscript to correct grammatical errors and addressed all missing citations to references and sections in the Appendix.
- We have revised the caption of Figure 1 to enhance clarity and comprehensiveness.We are also updating the entire figure but it did not get through in the updated version. We will include it in the final version of the paper.
- We have improved the quality of all figures throughout the manuscript and enhanced their captions to provide clearer and more detailed explanations.

---

> ### Author Response · Authors · 2024-11-29
> **Global Response on Strengthening Clarity, Presentation and Evaluations (Contd.)**
>
> **Updates to Experiments**
>
> - We have streamlined the explanation of the basic environments Section 4.1 and the process of converting them to their Altared versions in Appendix A.1 and A.2. To achieve this, we have relocated the details of the agent architecture to the Appendix and expanded the description of each environment. Each subsection is now self-contained, providing a clearer and more comprehensive overview for each environment. We would also like to bring details from Appendix A to the main paper but will need either an extra page or further reorganization.
>
> - We have addressed concerns regarding variance in the plots by running all experiments with additional seeds and extended training durations. For Commons Harvest, our results continue to demonstrate strong performance compared to the baseline without institutions. For Allelopathic Harvest, we observe improved performance, and, more importantly, reduced variance in our approach over training time. This reduction in variance indicates that agents adapt to dynamic norms more effectively with the inclusion of institutions in our framework.
> - We have refreshed all plots to enhance their visual quality and accessibility. Additionally, we have streamlined the discussion of qualitative results to improve clarity and coherence.
> - The missing code, as noted by a couple of reviewers, was an oversight on our part. It has now been made available. It is available at the same link: https://anonymous.4open.science/r/normarl-public-804C and the implementation of environment conditions can be viewed in the meltingpot/configs/substrates folder.
>
> **Beyond specific concerns, we emphasize that our work makes several fundamental contributions:**
>
> - Formally extends Markov games to incorporate a classification institution to study the impact of explicit normative infrastructure on alignment.
> - Provides a generalizable recipe and examples for converting a given Markov game into an altar-augmented setting
> - Demonstrates that the presence of explicit classification institutions, such as the altar, improves agents’ ability to achieve cooperation, enforce norms accurately, and adapt effectively to uncertainty.
> - Showcases the utility of multi-agent reinforcement learning (MARL) as a tool for investigating complex normative challenges, while emphasizing the critical importance of incorporating institutional structures into environments designed to simulate and study socio economic phenomena.
>
>
> [1] Punishment allows the evolution of cooperation (or anything else) in sizable groups,, Robert Boyd and Peter J Richerson, Ethology and sociobiology, 1992.

---

### Meta-Review · Area_Chair_7NN8 · 2024-12-21

**Metareview:**

The paper proposes Altared Games, a framework that extends Markov games by introducing an institutional feature called the “altar” to encode explicit normative rules for agents in multi-agent reinforcement learning (MARL). The study evaluates Altared Games in two mixed-motive environments—Commons Harvest and Allelopathic Harvest—demonstrating that explicit institutional representations improve agents’ ability to align with cooperative norms and achieve higher social welfare compared to environments without such features.

The main strengths of the work lie in its choice of topics and its innovative approaches to addressing the research questions. However, the initial reviews raised several concerns, mostly regarding the presentation (e.g., setup, figure descriptions, results, contributions). The authors provided extensive responses and submitted a heavily revised manuscript. Only one reviewer engaged during the author-reviewer discussion period, while two additional reviewers indicated their evaluations do not change much despite reviewing the responses after the discussion period.

After reviewing the author responses and revisions myself, one concern I have is that the changes made are so significant that another round of reviews may be warranted. In particular, the revisions include corrections to implementation errors in the original figures in the results sections. These updated figures appear different enough to merit a careful re-evaluation. While one could argue that the qualitative properties remain similar, the visual differences in the figures justify a closer review. Additionally, several sections underwent heavy revisions without clear markings to indicate the changes. (A side note: in this situation, I think it's better for the authors to explicitly highlight modifications in the revisions, e.g., using colors, as the currently provided revision makes it challenging for reviewers to fully comprehend the differences to engage in discussion within a limited time period).

Given the extent of these changes, I believe it is challenging for the reviewers to thoroughly assess the updated version within the discussion period in the conference format. Therefore, I recommend rejection but encourage the authors to build upon the commendable and substantial effort they have put into this revision for their next submission.

**Additional Comments On Reviewer Discussion:**

The authors provided extensive responses and submitted a significantly revised manuscript during the rebuttal. Only one reviewer participated in the author-reviewer discussion, while two others indicated that their evaluations remained unchanged after reviewing the responses. After reviewing the author responses and revisions myself, I believe the extent of the revisions (e.g., correcting implementation issues to update the figures in the result section and making substantial changes in several sections) warrants another round of reviews.

---

### Decision · Program_Chairs · 2025-01-22

Reject